# SPIRAL: Self-Play on Zero-Sum Games Incentivizes Reasoning via Multi-Agent Multi-Turn Reinforcement Learning

**Bo Liu\*[1], Simon Yu\*[2], Zichen Liu\*[1,3], Leon Guertler\*[4]**
**Penghui Qi[1,3], Daniel Balcells[5], Mickel Liu[6], Cheston Tan[4], Weiyan Shi[2], Min Lin[3], Wee Sun Lee[1]**
**Natasha Jaques[†6]**
[1]National University of Singapore    [2]Northeastern University    [3]Sea AI Lab
[4]Centre for Frontier AI Research (CFAR), A\*STAR    [5]Plastic Labs    [6]University of Washington

## ABSTRACT

Recent advances in reinforcement learning have shown that language models can develop sophisticated reasoning through training on tasks with verifiable rewards, but these approaches depend on human-curated problem-answer pairs and domain-specific reward engineering. We introduce SPIRAL, a self-play framework where models learn by playing **multi-turn, zero-sum games against continuously improving versions of themselves**, generating an automatic curriculum of stronger opponents, and eliminating the need for human supervision. To enable this self-play training at scale, we implement a fully online, multi-turn, multi-agent reinforcement learning system for LLMs and propose role-conditioned advantage estimation (RAE) to stabilize multi-agent training. SPIRAL produces reasoning capabilities that transfer broadly, improving performance by up to 10% across a suite of 8 reasoning benchmarks on 4 different models spanning Qwen and Llama model families, outperforming supervised fine-tuning on 25,000 expert game trajectories. Multi-game training (*TicTacToe*, *Kuhn Poker*, *Simple Negotiation*) yields the strongest results, with improvements observed across both base and instruction-tuned models. Analysis of chain-of-thought traces reveals that games develop distinct cognitive patterns that transfer to improve reasoning performance, with different games developing complementary strengths. Even models which have already been trained on reasoning tasks using RLVR, like DeepSeek-R1-Distill-Qwen-7B, still benefit from our approach. These results demonstrate that zero-sum games naturally develop transferable reasoning capabilities across diverse model architectures and training stages, highlighting a promising direction for autonomous reasoning development. Our code can be found in `https://github.com/spiral-rl/spiral`.

## 1 INTRODUCTION

Recent breakthroughs in language model reasoning, including OpenAI o1 (OpenAI, 2024) and DeepSeek-R1 (DeepSeek Team, 2024), reveal that reinforcement learning (RL) can unlock dramatic improvements in Chain-of-Thought reasoning (Wei et al., 2022). Through outcome-based rewards, RL enables models to develop generalizable reasoning strategies and consistently solve complex problems where supervised fine-tuning shows limited progress.

However, current approaches face a fundamental scalability bottleneck: dependence on carefully engineered reward functions, domain-specific datasets, and expert supervision (DeepSeek Team, 2024; Ouyang et al., 2022; Bai et al., 2022). Each new reasoning domain requires experts to craft evaluation metrics, curate training problems, and validate reasoning traces. This manual process becomes increasingly unsustainable as we pursue more general intelligence, limiting both scale and diversity of reasoning challenges that models can learn from.

---

\* Equal contribution, order randomly decided by dice roll.
† Corresponding author.

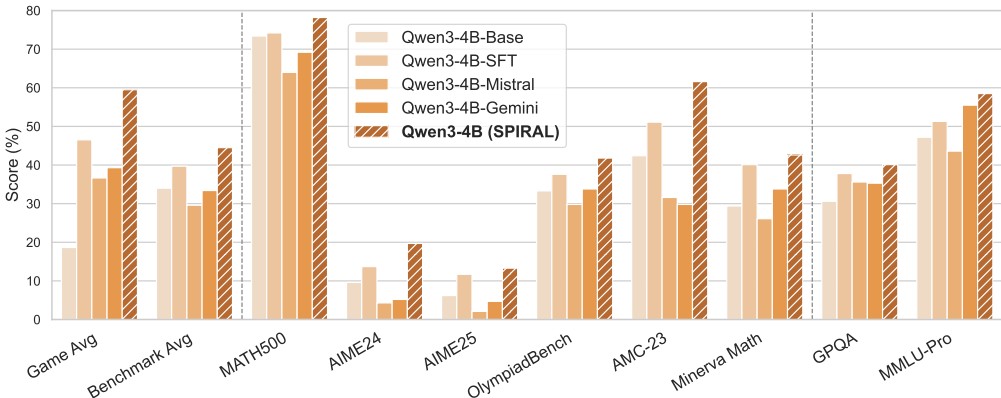

Figure 1: SPIRAL achieves consistent improvements over base models across game performance and reasoning benchmarks. It also surpasses SFT on expert game trajectories and RL baselines trained against fixed opponents (Mistral and Gemini).

Self-play on games offers a solution by eliminating human supervision in training data creation (Silver et al., 2017; Tesauro, 1995). In game-based self-play, models learn by competing against copies of themselves, where game outcomes provide automatic feedback and opponents improve equally, maintaining a consistent challenge that drives continuous learning. Although many prominent successes in past AI research relied on self-play—from TD-Gammon (Tesauro, 1995) to AlphaGo (Silver et al., 2016; 2017) to OpenAI Five (Berner et al., 2019)—so far, applying self-play on games to enhance language model reasoning remains largely unexplored. Prior attempts have been limited to simple word games with offline updates (Cheng et al., 2024), LoRA adaptations (Dettmers et al., 2023; Park et al., 2025), or single-turn tasks (Zhao et al., 2025), falling short of leveraging multi-turn competitive dynamics for extended strategic reasoning.

We introduce **SPIRAL (Self-Play on zero-sum games Incentivizes Reasoning via multi-Agent multi-turn reinforcement Learning)**, which applies self-play to two-player zero-sum language games for developing reasoning capabilities. SPIRAL offers two key advantages: unlike traditional RLVR approaches depending on human-curated problem-answer pairs, it generates unlimited training data through game dynamics alone; compared to fixed-opponent training (see Fig. 2), self-play prevents overfitting to static strategies by continuously evolving challenge level. However, implementing this for LLMs presents significant challenges. The computational demands of multi-turn, multi-agent autoregressive generation require sophisticated distributed systems, while standard RL algorithms suffer from high variance in multi-agent settings. We address these through a fully online, multi-turn, multi-agent reinforcement learning system with distributed actor-learner architecture and introduce role-conditioned advantage estimation (RAE), which stabilizes training by normalizing rewards relative to each player's expected performance.

**Key Findings.** Training on zero-sum games produces reasoning capabilities that transfer broadly across diverse model architectures. Multi-game SPIRAL training (TicTacToe, Kuhn Poker, Simple Negotiation) achieves up to 10% improvement across 8 reasoning benchmarks, outperforming supervised fine-tuning on 25,000 expert trajectories. On Qwen3-4B-Base (Yang et al., 2025), multi-game training reaches 44.5% average performance versus 34.0% baseline (+10.5% absolute gain), while Qwen3-8B-Base (Yang et al., 2025) improves from 39.5% to 49.6% (+10.1%). The approach generalizes across model families: base models (Qwen3-4B/8B-Base, Octothinker-8B-Base; Wang et al. (2025a)) and instruction-tuned models (Llama-3.1-8B-Instruct; Dubey et al. (2024)) all show consistent improvements, with Octothinker-8B-Base gaining 8.0% and Llama-3.1-8B-Instruct improving 2.0% despite already being instruction-tuned. Each game develops complementary cognitive skills: TicTacToe for spatial reasoning, Kuhn Poker for probabilistic thinking, and Simple Negotiation for strategic optimization, which combine synergistically in multi-game training. Using post-hoc analysis, we find examples of three patterns learned from gameplay that transfer to improve math performance: case-by-case analysis, expected value calculation, and pattern recognition. These patterns develop effectively through self-play's adaptive curriculum, as fixed-opponent training fails while self-play continuously improves. Role-conditioned Advantage Estimation proves critical: without

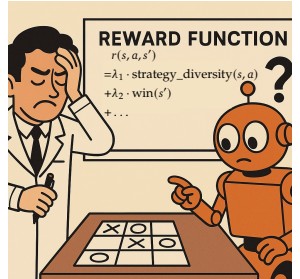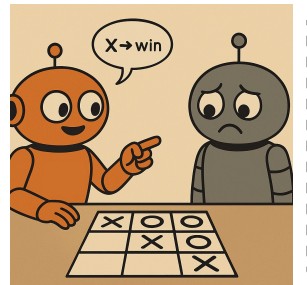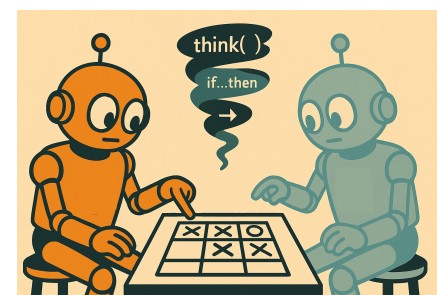

Figure 2: From human-designed rewards to self-discovered reasoning through SPIRAL. **Left**: Traditional RL requires human experts to design complex reward functions. **Middle**: Fixed opponent training leads to exploitation of static strategies. **Right**: SPIRAL enables continuous reasoning improvement through self-play, where both players develop increasingly sophisticated strategies without human supervision.

RAE, models abandon reasoning after 200 steps, progressively generating empty thinking traces that destroy generalization. Building on these findings, our work makes the following contributions:

1. **A Fully Online, Multi-Turn, Multi-Agent RL Framework for LLMs:** We develop a distributed actor-learner architecture enabling online self-play with full-parameter updates across multiple two-player zero-sum language games. The multi-turn aspect trains models to reason through sequential decisions, directly preparing them for complex multi-step problem solving. Unlike prior offline approaches, this provides continuous curriculum as the model adapts to an ever-improving opponent. We release our implementation to facilitate further research.

2. **Role-conditioned Advantage Estimation (RAE):** We introduce a variance-reduced advantage estimator specifically designed for multi-agent settings. By normalizing rewards relative to each player's expected performance, RAE prevents the degradation of the model's reasoning capabilities, a failure mode we term "thinking collapse". Without it, models progressively abandon reasoning traces after 200 steps, which is critical for generalization.

3. **Empirical Discovery of Transfer:** We demonstrate that self-play on zero-sum games improves both out-of-distribution game performance and academic reasoning benchmarks by up to 10% without domain-specific training data. Our analysis identifies reasoning patterns (systematic decomposition, expected value calculation, case-by-case analysis) that transfer from games to mathematics at measurable rates, with different games developing specialized skills that combine synergistically in multi-game training.

## 2 RELATED WORK

**Reinforcement Learning for LLM Reasoning.** Reinforcement learning (RL) in LLMs has progressed from alignment tasks using RLHF (Jaques et al., 2019; Ouyang et al., 2022; Bai et al., 2022) to directly improving reasoning capabilities. Recent models like OpenAI o1 (OpenAI, 2024) and DeepSeek-R1 (DeepSeek Team, 2024) demonstrate that RL with verifiable rewards (RLVR) can unlock chain-of-thought reasoning using rule-based rewards (Lightman et al., 2023; Uesato et al., 2022). However, these approaches depend on human-curated problem sets and domain-specific reward engineering. SPIRAL eliminates this dependency by using self-play games to generate unlimited reasoning challenges without human supervision.

**Self-Play and Multi-Agent RL for LLMs.** Self-play in LLMs initially focused on alignment objectives (Chen et al., 2024; Yuan et al., 2024) before recent work applied it to enhance model capabilities. SPAG (Cheng et al., 2024) applies self-play to Adversarial Taboo using offline updates on a single game; SPC (Chen et al., 2025) and Genius (Xu et al., 2025a) require predefined human task distributions; Absolute Zero (Zhao et al., 2025) generates single-turn coding tasks; Foundation Model Self-Play (Dharna et al., 2025) uses foundation models to evolve code-based policies rather than direct gameplay; Prover-Verifier Game (Kirchner et al., 2024) improve output legibility through adversarial training. Implementing multi-agent RL (MARL) for full-scale LLMs presents significant technical challenges (Wan et al., 2025; Liu et al., 2025b;a). Prior work circumvents these challenges by using RNNs instead of transformers (Sarkar et al., 2025), restricting to simpli-

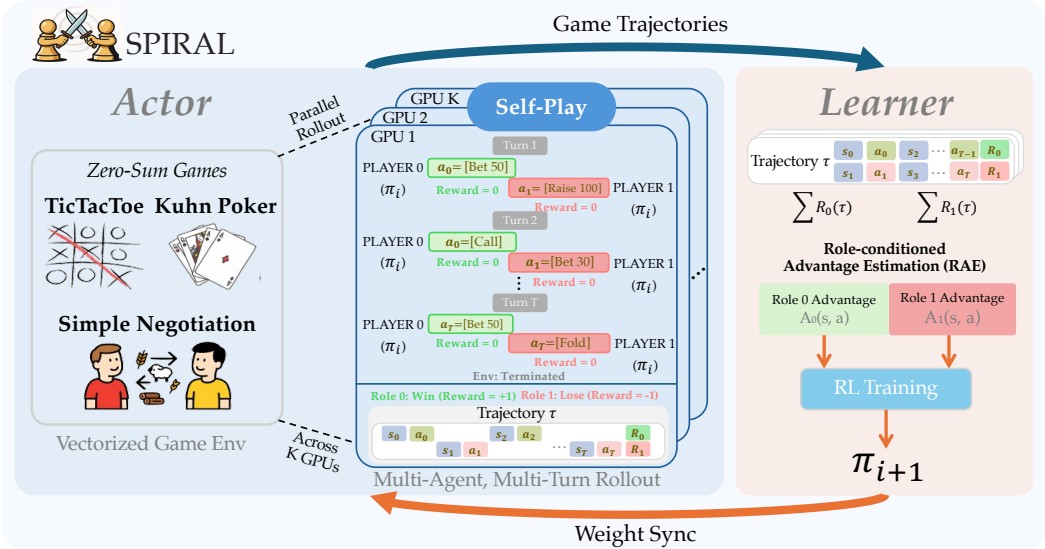

Figure 3: **The SPIRAL Framework.** SPIRAL employs an actor-learner architecture for scalable self-play training. Parallel actors sample trajectories from a diverse set of games using vectorized environments. A single policy $\pi_i$ plays both roles, generating zero-sum, sparse reward game trajectories. The centralized learner processes these trajectories using Role-conditioned Advantage Estimation (RAE) to compute separate advantages, $A_0(s, a)$ and $A_1(s, a)$, for each role. These are then used for on-policy reinforcement learning updates.

fied environments (Jacob et al., 2022; Sukhbaatar et al., 2018), or applying supervised fine-tuning on trajectories from proprietary models (Liao et al., 2024). In contrast, SPIRAL implements fully online, full-parameter MARL through a distributed actor-learner architecture, enabling continuous adaptation to evolving opponents across multiple games.

**LLMs in Gaming.** Games serve as both evaluation benchmarks (Paglieri et al., 2024; Ruoss et al., 2024; Zhang et al., 2024; Duan et al., 2024) and training domains (Feng et al., 2024; Verma et al., 2025). LMRL-Gym (Abdulhai et al., 2023) and RAGEN (Wang et al., 2025b) both employ single-agent multi-turn RL, with LMRL-Gym providing 8 benchmarking tasks and RAGEN focusing on trajectory-level optimization. ViGaL (Xie et al., 2025b) shows that single-agent RL on visual-spatial games transfers to mathematical reasoning without explicit math training. Logic-RL (Xie et al., 2025a) trains on synthetic puzzle games; Divide-Fuse-Conquer (Zhang et al., 2025) applies offline learning to grouped games; Boundless Socratic Learning (Schaul, 2024) uses language games for continual learning; Code2Logic (Tong et al., 2025) synthesizes reasoning data from game code. We also distinguish SPIRAL from works designed for achieving super-human performance at specific games. Cicero (FAIR et al., 2022) integrates a language model with a separate strategic planning algorithm to achieve human-level performance in Diplomacy. Similarly, agents developed for Werewolf (Xu et al., 2024; 2025b) and Avalon (Wang et al., 2023) focus on optimizing strategic communication and hidden-role deduction to win within their respective game rules. While these works target in-domain victory, SPIRAL treats competitive pressure as a training scaffold to develop reasoning patterns (e.g., case analysis) that transfer to out-of-domain reasoning tasks. SPIRAL uniquely combines three elements: (1) multi-agent self-play where both players share parameters, (2) fully online learning with continuous opponent evolution, and (3) demonstrated transfer from zero-sum language games to academic reasoning benchmarks achieving up to 10.5% improvement without exposure to benchmark-related problems during training.

## 3 THE SPIRAL FRAMEWORK

We present SPIRAL, a framework enabling language models to develop generalizable reasoning through multi-turn competitive self-play on games, illustrated in Figure 3.

**Formulation.** SPIRAL implements self-play through turn-based zero-sum language games from collection $\mathcal{G} = \{G_1, G_2, ..., G_n\}$. Each game $G_i$ is a two-player zero-sum Markov game (Littman,

1994) built on turn-level MDPs where states $s \in \mathcal{S}$ represent complete contexts (e.g., game configurations), actions $a \in \mathcal{A}$ are complete multi-token responses, and transition function $T_i$ determines state dynamics after full turn completion. The zero-sum property ensures $r_0(s, a^{(0)}, a^{(1)}) + r_1(s, a^{(0)}, a^{(1)}) = 0$ for all states and actions where $a^{(p)}$ denotes the action of player $p \in \{0, 1\}$, creating competitive dynamics. See Appendix C for detailed formulations.

**Benefits of multi-turn, zero-sum games.** Zero-sum dynamics create continuous improvement pressure through rewards given only at game termination: $r_i(s_t, a_t^{(0)}, a_t^{(1)}) = 0$ for all non-terminal states, with terminal rewards $R_0(\tau) = \rho_i(s_T)$ and $R_1(\tau) = -\rho_i(s_T)$ where $\rho_i : \mathcal{S}_i^{\text{terminal}} \to \{-1, 0, 1\}$ determines the outcome and $\tau$ represents the complete trajectory. This forces robust strategy development as models only receive feedback upon game completion. The multi-turn structure mirrors sequential reasoning problems: players alternate turns with $p = t \bmod 2$ acting at time $t$ while the opponent waits, training models to maintain context, plan ahead, and adapt strategies.

**Self-play.** Rather than training separate policies $\pi_{\theta_0}$ and $\pi_{\theta_1}$ for each player, SPIRAL uses a single shared policy $\pi_\theta$ with parameters $\theta$, setting $\theta_0 = \theta_1 = \theta$. Role conditioning through system prompts enables the model to learn distinct strategies for each position (see Appendix D.1). At each turn, the active player generates a complete response $y_t^{(p)} \sim \pi_\theta(\cdot | s_t, p, G_i)$ conditioned on current state $s_t$, player role $p$, and game $G_i$. From this response, we extract the action $a_t^{(p)}$ to update the game state via $s_{t+1} = T_i(s_t, a_t^{(0)}, a_t^{(1)})$ where $a_t^{(1-p)} = \emptyset$ for the inactive player. This shared-parameter approach ensures efficient use of GPU memory while also guaranteeing that as the model improves at one role, it simultaneously faces a stronger opponent, creating an automatic curriculum. Algorithm 1 presents the complete training procedure.

**RL objective.** To optimize this shared policy, we apply Monte Carlo policy gradient methods. Using REINFORCE (Williams, 1992), the gradient becomes:

$$\nabla_\theta J(\theta) = \mathbb{E}_{G \sim \mathcal{G}} \mathbb{E}_{\tau \sim \pi_\theta \times \pi_\theta | G} \left[ \sum_{t \in T_0} \nabla_\theta \log \pi_\theta(y_t^{(0)} | s_t, 0, G) \cdot R_0(\tau) + \sum_{t \in T_1} \nabla_\theta \log \pi_\theta(y_t^{(1)} | s_t, 1, G) \cdot R_1(\tau) \right],$$
(1)

where $T_p = \{t : t \bmod 2 = p\}$ denotes turns where player $p$ acted. This formulation uses Monte Carlo returns which suffer from high variance, particularly problematic in self-play where the opponent's strategy continuously evolves, making the environment non-stationary.

**Role-conditioned advantage estimation.** Self-play on zero-sum games implies using the same model to optimize for opposing objectives, since $R_1(\tau) = -R_0(\tau)$. This can lead to unstable training dynamics which impedes learning. To reduce the high variance inherent in multi-agent REINFORCE, we introduce Role-conditioned Advantage Estimation (RAE). In two-player games, even with a shared policy, different roles may have different expected returns due to game asymmetries (e.g., first-move advantage in TicTacToe, information asymmetry in Kuhn Poker). RAE maintains separate baselines $b_{G,p}$ for each game $G \in \mathcal{G}$ and role $p \in \{0, 1\}$, estimating expected return $\mathbb{E}[R_p(\tau)]$ for that role in that game. We update these baselines using exponential moving average with decay rate $\alpha \in [0, 1]$:

$$b_{G,p} \leftarrow \alpha b_{G,p} + (1 - \alpha) R_p(\tau), \quad A_{G,p}(\tau) = R_p(\tau) - b_{G,p}$$
(2)

This provides better variance reduction than a global baseline by accounting for role-specific asymmetries. The variance-reduced policy gradient becomes:

$$\nabla_\theta J_{\text{SPIRAL}}(\theta) = \mathbb{E}_{G \sim \mathcal{G}} \mathbb{E}_{\tau \sim \pi_\theta \times \pi_\theta | G} \left[ \sum_{p \in \{0,1\}} \sum_{t \in T_p} A_{G,p}(\tau) \cdot \nabla_\theta \log \pi_\theta(y_t^{(p)} | s_t, p, G) \right]$$
(3)

By centering returns around role-specific expectations, RAE ensures gradient updates reflect genuine learning signal rather than inherent positional advantages. We do not normalize by response length to avoid length bias (Liu et al., 2025c). The complete procedure is in Algorithm 1.

**Implementation.** To implement SPIRAL, we develop a truly online multi-agent, multi-turn RL system for finetuning LLMs. Our training framework builds on Oat (Liu et al., 2024), which provides interfaces of a distributed actor-learner architecture (Espeholt et al., 2018). We instantiate actors

---

**Algorithm 1** SPIRAL: Role-Balanced Multi-Turn Self-Play

---

**Require:** Policy $\pi_\theta$, Games $\mathcal{G} = \{G_1, ..., G_n\}$, decay rate $\alpha \in [0, 1]$
 1: Initialize baselines $b_{G_i,p} = 0$ for all $G_i \in \mathcal{G}, p \in \{0, 1\}$
 2: **while** not converged **do**
 3:     **// Self-Play Trajectory Collection**
 4:     $\mathcal{B} \leftarrow \emptyset$
 5:     **for** $k = 1$ to $K$ actors in parallel **do**
 6:         Sample game $G_i \sim \mathcal{G}$, initialize $s_0 \sim G_i$
 7:         **for** turn $t = 0, 1, 2, ...$ until terminal **do**
 8:             $p \leftarrow t \bmod 2$                                    $\triangleright$ Determine active player
 9:             $y_t^{(p)} \sim \pi_\theta(\cdot | s_t, p, G_i)$              $\triangleright$ Generate '[reasoning] $\boxed{[action]}$'
10:             $a_t^{(p)} \leftarrow \text{extract\_action}(y_t^{(p)})$
11:             $a_t^{(1-p)} \leftarrow \emptyset$                          $\triangleright$ Inactive player
12:             $s_{t+1} \leftarrow T_i(s_t, a_t^{(0)}, a_t^{(1)})$
13:         **end for**
14:         $R_0 \leftarrow \rho_i(s_T), R_1 \leftarrow -R_0$
15:         Define $\tau = \{(s_0, y_0), (s_1, y_1), \ldots, (s_T, y_T)\}$
16:         Add $(\tau, G_i)$ to batch $\mathcal{B}$                        $\triangleright$ Store trajectory with its game
17:     **end for**
18:     **// Role-Balanced Policy Optimization**
19:     **for** $(\tau, G_i) \in \mathcal{B}$ **do**
20:         **for** $p \in \{0, 1\}$ **do**
21:             $b_{G_i,p} \leftarrow \alpha b_{G_i,p} + (1 - \alpha) R_p(\tau)$
22:             $A_{G_i,p}(\tau) \leftarrow R_p(\tau) - b_{G_i,p}$
23:         **end for**
24:     **end for**
25:     Update $\theta$ on full sequences $y_t$ using REINFORCE with advantages $A_{G_i,p}$ (Eq. 3)
26: **end while**

---

to execute the self-play loop, using vLLM (Kwon et al., 2023) for efficient model inference and TextArena (Guertler et al., 2025) to simulate the language games. The resulting multi-turn, multi-game self-play experiences are used to update the LLM via policy gradient methods (Sutton & Barto, 2018), incorporating our proposed Role-conditioned Advantage Estimation in the collocated learner.

## 4 EXPERIMENTAL RESULTS

We evaluate SPIRAL across diverse model architectures and game environments to understand how self-play develops transferable reasoning capabilities. We train on three games from TextArena (Guertler et al., 2025): TicTacToe (spatial reasoning), Kuhn Poker (probabilistic reasoning), and Simple Negotiation (strategic optimization). Models include Qwen3-4B/8B-Base (Yang et al., 2025), Llama-3.1-8B-Instruct (Dubey et al., 2024), and Octothinker-8B-Base (Wang et al., 2025a). Training spans 400 steps with 128 samples per step on 8 H100 GPUs, using Adam optimizer with learning rate $1 \times 10^{-6}$ and temperature 1.0. We evaluate on eight reasoning benchmarks (MATH500, OlympiadBench, Minerva Math, AIME24/25, AMC23, GPQA-Diamond, MMLU-Pro) and seven out-of-distribution games. Complete implementation details are in Appendix D.

**Self-play on games transfers to improve reasoning.** The central results of this paper are shown in Table 1, which demonstrates that multi-game SPIRAL training achieves up to 10.5% improvement on reasoning benchmarks without domain-specific data. with Qwen3-4B-Base improving from 34.0% to 44.5% average performance (+10.5%). We also compare with supervised fine-tuning (SFT) on 25,000 expert game trajectories, generated by Qwen3-32B models, which improves performance on several benchmarks including AIME24 and AIME25, revealing that games themselves contain skills relevant to reasoning. However, SPIRAL consistently outperforms SFT across all 8 benchmarks, demonstrating that self-play discovers more effective reasoning strategies than imitating expert demonstrations.

Table 1: Reasoning benchmark performance. The "-Kuhn" suffix denotes fine-tuning solely on a single game (Kuhn Poker), while the "-Multi" suffix indicates fine-tuning on all three games. SPIRAL improves reasoning without any domain-specific training data. *Few-shot evaluation following Qwen3 technical report.

| Model | Math500 | AIME24 | AIME25 | Olympiad | AMC-23 | Minerva | GPQA-D | MMLU-Pro | Average |
|---|---|---|---|---|---|---|---|---|---|
| **Qwen3-4B-Base** | 73.4 | 9.6 | 6.2 | 33.3 | 42.4 | 29.4 | 30.6* | 47.2* | 34.0 |
| + SFT-Kuhn | 74.0 | 11.0 | 10.4 | 36.7 | 48.6 | 36.8 | 33.0 | 48.8 | 37.4 |
| + SFT-Multi | 74.2 | 13.7 | 11.7 | 37.6 | 51.1 | 40.1 | 37.8 | 51.3 | 39.7 |
| + SPIRAL-Kuhn (Ours) | 76.4 | 18.2 | **15.6** | 38.4 | 61.2 | 42.4 | 37.0 | 57.7 | 43.4 |
| + SPIRAL-Multi (Ours) | **78.2**$_{+4.8}$ | **19.7**$_{+10.1}$ | 13.3$_{+7.1}$ | **41.8**$_{+8.5}$ | **61.6**$_{+19.2}$ | **42.6**$_{+13.2}$ | **40.1**$_{+9.5}$ | **58.5**$_{+11.3}$ | **44.5**$_{+10.5}$ |
| **Qwen3-8B-Base** | 77.0 | 12.1 | 11.2 | 33.5 | 50.6 | 38.2 | 38.0* | 55.7* | 39.5 |
| + SFT-Multi | 82.8 | 19.9 | 15.6 | 45.9 | 63.5 | 40.8 | 41.6 | 58.8 | 46.1 |
| + SPIRAL-Multi (Ours) | **86.6**$_{+9.6}$ | **26.2**$_{+14.1}$ | **16.8**$_{+5.6}$ | **49.6**$_{+16.1}$ | **65.2**$_{+14.6}$ | **46.3**$_{+8.1}$ | **44.6**$_{+6.6}$ | **61.1**$_{+5.4}$ | **49.6**$_{+10.1}$ |
| **Octothinker-8B-Base** | 65.6 | 1.7 | 0.5 | 26.6 | 33.5 | 25.7 | 22.1 | 30.8 | 25.8 |
| + SFT-Multi | 66.0 | 3.3 | 3.8 | 23.9 | 31.0 | 23.8 | 24.9 | 39.1 | 27.0 |
| + SPIRAL-Multi (Ours) | **68.6**$_{+3.0}$ | **5.3**$_{+3.6}$ | **4.8**$_{+4.3}$ | **33.7**$_{+7.1}$ | **43.2**$_{+9.7}$ | **32.0**$_{+6.3}$ | **33.8**$_{+11.7}$ | **49.3**$_{+18.5}$ | **33.8**$_{+8.0}$ |
| **Llama-3.1-8B-Instruct** | 46.4 | 4.6 | 0.7 | 13.8 | 23.3 | 22.8 | 30.2 | 49.1 | 23.9 |
| + SFT-Multi | **51.8** | 4.6 | 0.7 | **19.1** | 23.3 | 21.7 | 30.0 | 48.9 | 25.0 |
| + SPIRAL-Multi (Ours) | 49.8$_{+3.4}$ | **4.9**$_{+0.3}$ | **1.8**$_{+1.1}$ | 17.3$_{+3.5}$ | **26.0**$_{+2.7}$ | **24.6**$_{+1.8}$ | **32.2**$_{+2.0}$ | **50.4**$_{+1.3}$ | **25.9**$_{+2.0}$ |
| **DeepSeek-Distill-Qwen-7B** | 90.8 | 53.0 | 39.5 | 56.9 | **89.3** | 48.2 | 48.6 | 57.1 | 60.4 |
| + SFT-Multi | 91.8 | 49.3 | 36.6 | 52.4 | 88.2 | 48.2 | 44.5 | 55.6 | 58.3 |
| + SPIRAL-Multi (Ours) | **93.0**$_{+2.2}$ | **54.1**$_{+1.1}$ | **40.8**$_{+1.3}$ | **57.9**$_{+1.0}$ | **89.3**$_{+0.0}$ | **51.1**$_{+2.9}$ | **49.6**$_{+1.0}$ | **58.9**$_{+1.8}$ | **61.8**$_{+1.4}$ |

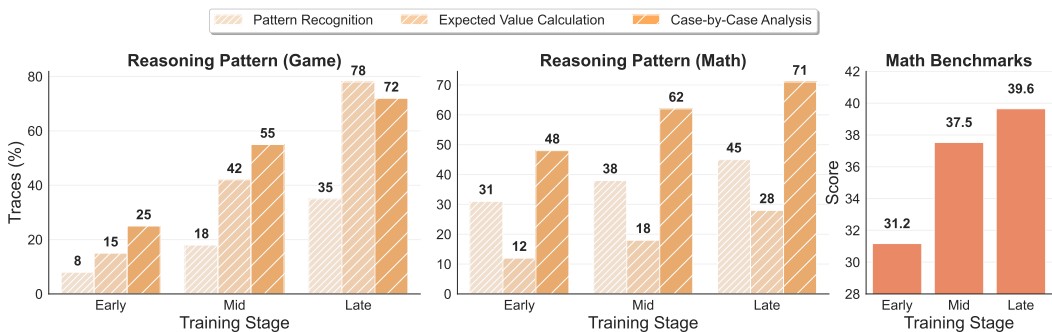

Figure 4: **Evolution of reasoning patterns during SPIRAL training and their transfer to mathematical reasoning.** We track three core reasoning patterns (Pattern Recognition, Expected Value Calculation, and Case-by-Case Analysis) across 290 game trajectories and 46,792 math solutions. *Left*: Patterns increase via training on games, with Expected Value Calculation reaching 78% by late training. *Middle*: These patterns transfer to mathematical reasoning with varying effectiveness: Case-by-Case Analysis maintains high transfer (72% to 71%), Pattern Recognition shows amplification (35% to 45%), while Expected Value Calculation transfers more selectively (78% to 28%). *Right*: Math benchmark scores improve from 31.2 to 39.6 as these reasoning patterns develop, demonstrating that game-learned strategies enhance mathematical problem-solving capabilities.

## 4.1 UNDERSTANDING WHY SPIRAL WORKS

**Emergent reasoning patterns learned on games transfer to math questions.** To understand how games can improve reasoning performance, we analyzed chain-of-thought patterns using GPT-4.1 (OpenAI, 2025) to classify 290 game trajectories and 46,792 math solutions. Table 2 illustrates three core patterns that emerge during gameplay and transfer to mathematics. Figure 4 tracks their evolution. In the late training stage, the transfer from games to math is clear: Case-by-Case Analysis transfers near-perfectly (72% to 71%), Pattern Recognition amplifies (35% to 45%), while Expected Value Calculation transfers selectively (78% to 28%). The success of this transfer may stem from two factors. This transfer likely works because evolving opponents prevent memorization while games teach pure reasoning strategies without requiring mathematical knowledge.

Table 2: Representative reasoning patterns and their manifestations in games and mathematics. We use an LLM-as-a-judge framework to evaluate the frequency of these patterns over training.

| Pattern | Game Context (Kuhn Poker) | Math Transfer |
|---|---|---|
| **Case-by-Case Analysis** | `<output>` **Case 1 - Fold:** You lose 1 chip. You have 2 chips now. **Case 2 - Call:** You have 0% chance of winning, so you will have 1 chip if you lose. Since losing 1 chip (fold) is better than losing 2 chips (call and lose), the best action is to $\boxed{fold}$. `</output>` | `<output>` To solve $m^2 + 2 \cdot 3^n = m(2^{n+1} - 1)$: **Case $n = 0$:** $m^2 + 2 = m$, no real solutions **Case $n = 1$:** $m^2 + 6 = 3m$, no real solutions **Case $n = 2$:** $m^2 + 18 = 7m$, no real solutions **Case $n = 3$:** $m^2 + 54 = 15m$ gives $m = 6, 9$ (valid) **Case $n = 4$:** $m^2 + 162 = 31m$, no integer solutions **Case $n = 5$:** $m^2 + 486 = 63m$ gives $m = \boxed{9, 54}$ (valid) `</output>` |
| **Expected Value Calculation** | `<output>` With card J against opponent's bet: **EV(call)** = P(win) × 2 - P(lose) × 2 = 0 × 2 - 1 × 2 = -2 **EV(fold)** = -1 (lose ante) Since EV(fold) > EV(call), $\boxed{fold}$ is optimal. `</output>` | `<output>` For average of $A + BC + DEF$ over permutations: **E[A]** = $\frac{\sum a_i}{6} = \frac{-2}{6} = -\frac{1}{3}$ **E[BC]** = $(E[B])(E[C]) = (-\frac{1}{3})^2 = \frac{1}{9}$ **E[DEF]** = $(E[D])(E[E])(E[F]) = (-\frac{1}{3})^3 = -\frac{1}{27}$ Total $= -\frac{1}{3} + \frac{1}{9} - \frac{1}{27} = \boxed{-\frac{7}{27}}$ `</output>` |
| **Pattern Recognition** | `<output>` Previous rounds: Player 0 had K both times and won. **Pattern identified**: Player 0 likely has strong cards or bluffs consistently. Given I have J (weak), $\boxed{betting}$ might exploit their aggressive calling pattern. `</output>` | `<output>` Sum of three consecutive integers $(n-1) + n + (n+1) = 3n$. **Pattern recognized**: sum is always divisible by 3. For perfect cube: $3n = k^3$, so $k$ must be divisible by 3. Smallest: $k = \boxed{3} \Rightarrow 3n = 27$ `</output>` |

**Adaptive curriculum beats static opponents.** Self-play creates an automatic curriculum that adapts to model capabilities. Figure 5 compares self-play against fixed opponents (Random, Mistral-Small-3, Gemini-2.0-Flash-Lite[1]). Random opponents cause collapse: although they provide randomized rewards with positive expected value, similar to spurious rewards that might upweight certain base model behaviors (Shao et al., 2025), we observe this is insufficient to improve performance. Fixed model opponents enable initial learning but plateau once exploitable strategies are found. That fixed opponents like Gemini yield smaller gains reveals the effects are not merely from learning game mechanics or spurious rewards (Shao et al., 2025), but specifically from the adaptive curriculum. Unlike static baselines, self-play's continuously evolving challenge forces genuine reasoning development rather than exploitation of static patterns. Table 3 confirms this: self-play maintains 50-52% win rates while fixed-opponent training rises from 0% to 62.5%, indicating exploitation rather than continued learning.

Table 3: Win rates at different training stages of *Gemini Opponent* and *Self-Play* vs its opponent.

| Training Stage | Gemini Opponent Win Rate vs Gemini-2.0-Flash-Lite | Self-Play Win Rate vs Self (t-16) |
|---|---|---|
| Step 16 | 0.0% | 52.3% |
| Step 128 | 37.5% | 51.7% |
| Step 384 | 62.5% | 50.9% |

**Different games develop complementary skills.** Each game cultivates distinct cognitive abilities that transfer to related domains; specifically, we selected TicTacToe to target spatial reasoning, Kuhn Poker for probabilistic inference, and Simple Negotiation for strategic optimization. Table 4 tests how well agents trained on a specific game ('*specialists*') transfer to novel out-of-distribution (OOD) games. We find specialists transfer effectively to similar out-of-distribution games: TicTacToe specialists achieve 56.0% on Snake (spatial), Poker specialists dominate Pig Dice at 91.7% (probabilistic), Negotiation specialists win 55.8% on Truth and Deception (strategic). Multi-game training combines these skills synergistically, as shown in Table 5, which shows win-rate against Gemini-2.0-Flash[2]. Multi-game agents achieve 59.5% average performance, outperforming all single-game

---

[1]Accessed via `https://openrouter.ai/google/gemini-2.0-flash-lite-001`.
[2]Accessed via `https://openrouter.ai/google/gemini-2.0-flash-001`.

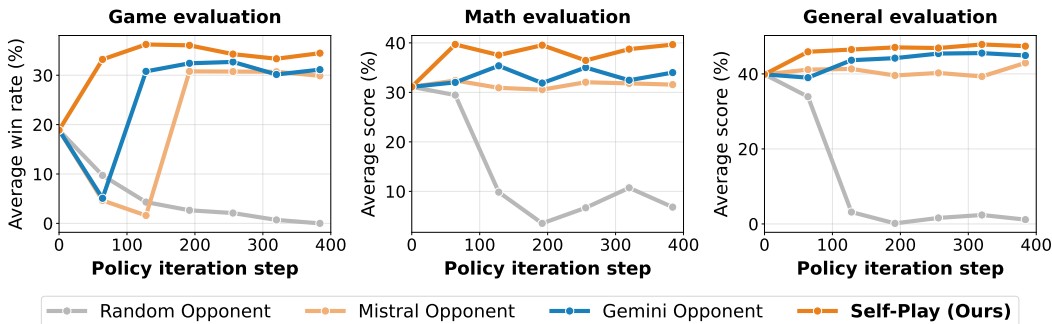

Figure 5: Performance comparison of self-play training and fixed-opponent baselines. All evaluations are averaged over multiple games/benchmarks (see App. D.3). Mistral Opponent refers to against Mistral-Small-3; Gemini Opponent refers to against Gemini-2.0-Flash-Lite.

specialists (best: 52.9%), using a fixed budget of 400 training steps determined by convergence analysis.

Table 4: Game specialists excel at both their training games and unseen games requiring similar cognitive skills. Each cell shows the win rate in head-to-head competition between specialists (e.g., 57.5% means TicTacToe specialist wins 57.5% of games against the other two specialists on TicTacToe). Bold indicates best performance in each column.

| Model | Training Games | | | OOD Games (Similar Skills) | | |
| | TicTacToe | Kuhn Poker | Simple Negotiation | Snake *(Spatial)* | Pig Dice *(Probabilistic)* | Truth and Deception *(Strategic)* |
| --- | --- | --- | --- | --- | --- | --- |
| TicTacToe Specialist | **57.5%** | 45.1% | 30.4% | **56.0%** | 56.7% | 48.7% |
| Poker Specialist | 45.5% | **64.2%** | 37.7% | 42.5% | **91.7%** | 45.4% |
| Negotiation Specialist | 40.5% | 40.2% | **62.7%** | 41.0% | 1.1% | **55.8%** |

Table 5: Multi-game training achieves competitive performance across all training games while excelling at novel composite challenges. All win rates shown are against Gemini-2.0-Flash as a fixed opponent. The multi-game model outperforms all specialists on average, demonstrating that diverse game training develops more flexible reasoning.

| Model | Training Games | | | OOD Games | | | |
| | TicTacToe | KuhnPoker | Simple Negotiation | Snake | Pig Dice | Truth and Deception | Average |
| --- | --- | --- | --- | --- | --- | --- | --- |
| Base Model | 17.5% | 21.5% | 15.6% | 7.8% | 0.2% | 49.6% | 18.7% |
| Random Policy | 24.5% | 31.3% | 8.2% | N/A | N/A | N/A | N/A |
| Instruct Model | 52.7% | 48.5% | 46.2% | 25.2% | 97.6% | 75.5% | 57.6% |
| **Single-Game Specialists** | | | | | | | |
| TicTacToe Specialist | **56.6%** | 24.4% | 30.5% | 28.1% | 97.6% | 79.9% | 52.9% |
| Kuhn Poker Specialist | 31.0% | 48.5% | 28.7% | 27.7% | 98.8% | 81.6% | 52.7% |
| Simple Negotiation Specialist | 27.7% | 16.8% | **39.1%** | 26.4% | 98.6% | 82.8% | 48.6% |
| **Multi-Game Model** | 54.3% | **53.9%** | 33.2% | **31.6%** | **99.8%** | **84.0%** | **59.5%** |

**Role-conditioned Advantage Estimation proves essential for stable training.** Figure 6 shows that without RAE, models suffer catastrophic thinking collapse within 100 policy iterations-reasoning trace lengths plummet from ~2,000 to near-zero characters, with models generating degenerate outputs like \boxed{bet}. General reasoning performance correspondingly drops from 44% to 40%. In contrast, RE-INFORCE with RAE maintains

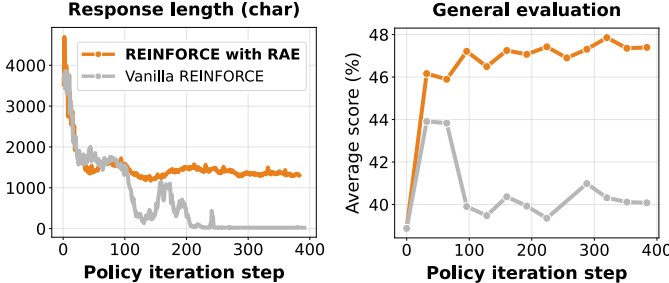

Figure 6: Training dynamics comparing REINFORCE with RAE (orange) versus vanilla REINFORCE (gray). RAE maintains stable performance across all metrics while vanilla REINFORCE suffers catastrophic thinking collapse. **Left**: Response length reveals thinking collapse where models stop generating reasoning traces; **Right**: Performance on general reasoning benchmarks.

stable response lengths around
1,300-1,500 characters and
improves performance from 40% to 47%. RAE achieves this stability by centering returns around role-specific baselines, preventing gradient variance from driving policies toward degenerate solutions.

## 5  CONCLUSION

We introduced SPIRAL, enabling language models to develop reasoning capabilities through competitive self-play without human-curated data. Our technical contributions include a fully online multi-turn MARL system for LLMs and Role-conditioned Advantage Estimation (RAE), which prevents thinking collapse in zero-sum games. Empirically, multi-game SPIRAL training improves reasoning benchmarks by up to 10.5% across diverse model architectures, surpassing supervised fine-tuning on 25,000 expert game trajectories. Different games develop distinct transferable skills (spatial, probabilistic, strategic) that combine synergistically. Analysis reveals that competitive gameplay forces discovery of reasoning patterns (case-by-case analysis, expected value calculation, pattern recognition) that transfer to academic domains.

SPIRAL demonstrates that simple games can unlock complex reasoning without domain-specific data. Future work could expand to cooperative games, incorporate partial observability, and design games targeting specific reasoning weaknesses. Understanding game-skill mappings could enable principled environment design for autonomous reasoning development.

## ACKNOWLEDGEMENTS

We thank Xidong Feng and Runji Lin for their helpful discussions and support throughout this project. We thank John Schulman for insightful feedback and encouragement. We would like to thank Thinking Machine Lab and Modal Lab for providing compute credits that supported our experiments. We also thank the TextArena team for maintaining the game environments used in this work. This research was supported by the Cooperative AI Foundation, the UW-Amazon Science Gift Hub, Sony Research Award, UW-Tsukuba Amazon NVIDIA Cross Pacific AI Initiative (XPAI), the Microsoft Accelerate Foundation Models Research Program, Character.AI, DoorDash, and the Schmidt AI2050 Fellows program. This material is based upon work supported by the Defense Advanced Research Projects Agency and the Air Force Research Laboratory, contract number(s): FA8650-23-C-7316. Any opinions, findings and conclusions, or recommendations expressed in this material are those of the author(s) and do not necessarily reflect the views of AFRL or DARPA.

## REPRODUCIBILITY STATEMENT

We provided the experiment code at https://github.com/spiral-rl/spiral. We have also provided the training settings in the experiment section (§4) and Appendix D. The experiments are run with an 8 H100 GPU cluster, and all calls for proprietary LLMs are via the official API or OpenRouter[3].

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

This appendix provides comprehensive details supporting our main findings. App. A documents the use of large language models in our analysis. App. B discusses limitations of our approach including reliance on designed game environments, computational requirements, evaluation constraints, and potential reward hacking risks. App. C provides the detailed formulations of turn-level MDPs and two-player zero-sum Markov games referenced in the main paper, showing how SFT and RLVR adapt to these frameworks. App. D presents complete implementation details including game environment observations, hyperparameter configurations, and evaluation settings for all benchmarks. App. E provides extended benchmark results across multiple base models, an extended analysis of our RAE ablation study with additional training dynamics, and a detailed case study showing the evolution of case-by-case analysis in mathematical problem-solving. App. F describes our systematic bottom-up approach for discovering and quantifying reasoning pattern transfer, including our GPT-4.1-assisted analysis framework. Finally, App. G specifies all game environments, detailing both training games (TicTacToe, Kuhn Poker, Simple Negotiation) and out-of-distribution evaluation games (Snake, Pig Dice, Truth and Deception).

## A LARGE LANGUAGE MODEL USAGE

We used large language models (LLMs) only for language refinement tasks, including grammar checking, phrasing adjustments, and enhancing readability. We also used Deep Research (OpenAI, 2025) to assist with related work search. Besides these, all scientific ideas, experiments, analyses, and results are the sole contributions of the authors.

## B LIMITATIONS

Our study, while promising, has several limitations that offer avenues for future research.

**Reliance on Designed Game Environments**   A core limitation is the dependency on engineered game environments. Although SPIRAL eliminates the need for human-curated problem datasets, it shifts the dependency to well-designed games. The games used in our experiments, such as the Tic-tactoe and Khun Poker, are relatively simple and feature dense rewards. It is an open question how well this approach **scales to more complex, open-ended environments** with sparse rewards, such as Minecraft or realistic robotics simulations. The design of the game environment itself may implicitly encode biases or heuristics that influence the agent's learned reasoning strategies, potentially limiting their generality.

**Computational Cost and Scalability**   The computational requirements for training are substantial. Each experimental run demanded **8 H100 GPUs for approximately 25 hours**, which may be prohibitive for many research groups. Furthermore, we observed that performance gains began to plateau after extended training periods. This suggests that simply scaling up the training duration with the current framework may yield diminishing returns, and more efficient algorithms or architectural improvements are necessary for further progress.

**Evaluation and Transferability**   Our evaluation, while comprehensive, has two key constraints:

- **Focus on Academic Benchmarks:** We primarily assessed reasoning on established academic benchmarks like MATH and GPQA. These benchmarks are excellent for measuring formal and scientific reasoning but do not capture the full spectrum of self-play.

- **Zero-Shot Evaluation:** The strict zero-shot evaluation setting tests for direct transfer but may not fully reveal the model's potential. Fine-tuning on a small set of target domain examples could potentially unlock significantly better performance, a possibility not explored in this work.

**Potential for Reward Hacking**   Like many reinforcement learning systems, SPIRAL is susceptible to **reward hacking**. An agent might discover policies that maximize the in-game score without learning the intended underlying reasoning skill. For instance, it could exploit a bug in the game physics or find a repetitive, degenerate strategy that succeeds for a narrow set of problems. While

we did not observe significant instances of this, it remains a risk, especially in more complex environments where robust reward shaping is challenging.

## C  PRELIMINARIES

This section provides the mathematical foundations and formal definitions underlying the SPIRAL framework, including turn-level MDPs, two-player zero-sum Markov games, and how existing training paradigms adapt to these formulations.

### C.1  TURN-LEVEL MARKOV DECISION PROCESSES (MDPS).

Language model training traditionally formulates generation as a token-level MDP (Bellman, 1957; Rafailov et al., 2024) where each action is a single token from vocabulary $\mathcal{V}$. For multi-turn reasoning and game-playing, we instead adopt a turn-level MDP formulation $\mathcal{M} = (\mathcal{S}, \mathcal{A}, T, r, \gamma)$. Here, states $\mathcal{S}$ represent complete contexts (e.g., game configurations, problem states, or conversation histories), actions $\mathcal{A}$ are complete responses (containing many tokens), the transition function $T : \mathcal{S} \times \mathcal{A} \to \Delta(\mathcal{S})$ determines state dynamics, $r : \mathcal{S} \times \mathcal{A} \to \mathbb{R}$ provides immediate rewards, and $\gamma \in [0,1]$ is the discount factor. The return is defined as the discounted sum of rewards: $R(\tau) = \sum_{t=0}^{T} \gamma^t r_t$.

The key distinction: in token-level MDPs, each decision outputs one token; in turn-level MDPs, each decision produces a complete multi-token response before transitioning. At each turn $t$, the language model observes state $s_t$ and generates:

$$y_t = \langle\text{think}\rangle c_t \langle/\text{think}\rangle \langle\text{answer}\rangle a_t \langle/\text{answer}\rangle, \tag{4}$$

where $c_t$ externalizes reasoning and $a_t \in \mathcal{A}$ is the executable action. (See App. C for how existing SFT and RLVR paradigms adapt to turn-level MDPs.)

### C.2  TWO-PLAYER ZERO-SUM MARKOV GAMES.

We extend the single-agent MDP to competitive settings with a two-player zero-sum Markov game (Littman, 1994) $\mathcal{G} = (\mathcal{S}, \mathcal{A}_0, \mathcal{A}_1, T, r, \gamma)$, where $\mathcal{A}_0$ and $\mathcal{A}_1$ are the action spaces for player 0 and player 1 respectively. The zero-sum property requires:

$$r_0(s, a^{(0)}, a^{(1)}) + r_1(s, a^{(0)}, a^{(1)}) = 0 \quad \forall s, a^{(0)}, a^{(1)}, \tag{5}$$

where $a^{(0)} \in \mathcal{A}_0$ and $a^{(1)} \in \mathcal{A}_1$ denote actions taken by each player. Given trajectory $\tau = \{(s_t, a_t^{(0)}, a_t^{(1)})\}_{t=0}^{T}$, the returns satisfy $R_1(\tau) = -R_0(\tau)$.

### C.3  SUPERVISED FINE-TUNING (SFT) IN TURN-LEVEL MDPS

In the turn-level setting, SFT requires a dataset $\mathcal{D}_{\text{SFT}} = \{(s_i, c_i^*, a_i^*)\}_{i=1}^{N}$ of states with expert reasoning traces $c_i^*$ and actions $a_i^*$. The model learns to imitate complete turn-level responses:

$$\mathcal{L}_{\text{SFT}}(\theta) = -\mathbb{E}_{(s, c^*, a^*) \sim \mathcal{D}_{\text{SFT}}} \left[\log \pi_\theta(c^*, a^*|s)\right]. \tag{6}$$

Note that in single-turn settings where each state $s$ appears only once, SFT reduces to standard behavior cloning. The key limitation remains: SFT requires expensive human annotation of both reasoning traces and final answers.

### C.4  REINFORCEMENT LEARNING WITH VERIFIABLE REWARDS (RLVR) IN TURN-LEVEL MDPS

RLVR (DeepSeek Team, 2024) eliminates the need for reasoning supervision, requiring only state-answer pairs $\mathcal{D}_{\text{RLVR}} = \{(s_i, a_i^*)\}_{i=1}^{N}$. In the turn-level formulation:

$$J_{\text{RLVR}}(\theta) = \mathbb{E}_{s \sim \mathcal{D}_{\text{RLVR}}, y \sim \pi_\theta(\cdot|s)} \left[r(s, a)\right], \tag{7}$$

where $r(s, a) = \mathbb{I}[a = a^*]$ indicates answer correctness and $y$ contains both reasoning and action.

In single-turn settings without subsequent interactions, RLVR reduces to a contextual bandit problem (Langford & Zhang, 2007). Recent works on mathematical (Shao et al., 2024; DeepSeek Team, 2024) and code reasoning (Zhu et al., 2024; Xin et al., 2024) show that even this simplified bandit-style RLVR can unlock sophisticated reasoning. However, these approaches still require human-curated problem sets $\mathcal{D}_{\mathrm{RLVR}}$, which SPIRAL eliminates through self-play.

# D   EXPERIMENTAL SETUP DETAILS

This section provides complete implementation details for reproducing our experiments. We begin with visual examples of game environments, followed by our hyperparameter configurations.

## D.1   GAME ENVIRONMENT OBSERVATIONS

The language models receive structured text observations from each game environment. Fig. 7 shows example observations from our three training games: TicTacToe, Kuhn Poker, and Simple Negotiation. These observations serve as the input prompts $s_t$ at each turn, providing complete game state information in natural language format.

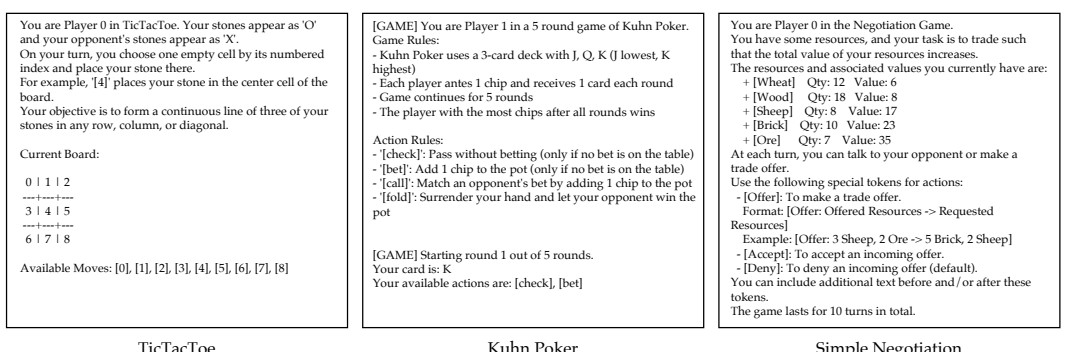

Figure 7: Example observations of three training game environments.

For games with partial observability such as Kuhn Poker and Simple Negotiation, we maintain Markovian state representations by concatenating historical actions into the current state $s_t$. This ensures the model has sufficient information for decision-making despite hidden information.

Similarly, Fig. 8 presents observations from five evaluation environments used to test out-of-distribution generalization. These games were never seen during training, allowing us to assess whether learned skills transfer to novel game mechanics.

## D.2   HYPERPARAMETER CONFIGURATION

Table 6 presents the complete hyperparameter settings used across all experiments. These configurations were selected through preliminary experiments to balance training stability and computational efficiency. Please see our open-source codebase for a complete and reproducible experiment example.

These hyperparameters remain fixed across all game environments and model scales to ensure fair comparison. The distributed training infrastructure utilizes 8 H100 GPUs, with parallel actors generating game trajectories while a centralized learner performs synchronous policy updates. On this hardware setup, the compute budget per experiment was approximately 25 hours for Qwen3-4B, 28 hours for Qwen3-8B, and 16 hours for both Llama3.1-8B-Instruct and Octothinker-8B. For SFT, we use Qwen3-32B (Yang et al., 2025) models to self-play on the targeted games, generating 25k winning trajectories and crafting them as the SFT dataset.

```
[GAME] 2-Player Snake on a 5×5 grid.
You control snake 0. Valid moves:
'[up]'/'[down]'/'[left]'/'[right]' (or w/s/a/d).
Objective: survive longest or be the longest and get the
highest score (turn limit 40 turns).
[GAME] Current Board:
+-----------+
| . . . . . |
| . 0 . A . |
| . . . . . |
| . . . 1 . |
| . . . . A |
+-----------+
```

```
[GAME] You are Player 0 playing a game of Pig Dice.
Rules:
- On your turn, you can either '[roll]' or '[hold]'
- Roll a 2-6: Add to your turn total
- Roll a 1: Lose turn total and end turn
- Hold: Add turn total to your score and end turn
- First to 100 points wins

When it's your turn, you'll see the current scores and turn
total.
Respond with '[roll]' to roll the die or '[hold]' to bank your
points.
[GAME] You current turn total is 0.
This is the first roll of your turn.
Available actions: '[roll]' or '[hold]'
```

```
[GAME] You are Player 0, and you are the Deceiver in the
'Truth and Deception' game.
You know which of the following two facts is correct:
Fact 1: The blue-ringed octopus is harmless to humans.
(wrong)
Fact 2: The blue-ringed octopus is one of the most
venomous marine animals. (correct)
Your goal is to convince the other player to choose the
incorrect fact.
You will have a conversation for 6 turns.
After that, the other player will make their guess.
On your turn, simply type your message.
```

Snake — Pig Dice — Truth And Deception

```
[GAME] You are Player 0 in an 2-player Liar's Dice game.
Rules:
- On your turn, you may either:
  1) Make a new bid with a higher quantity or higher face
(or both) than the current bid; i.e. '[Bid: 3, 4]',
  2) Call the last bid by typing '[Call]'.

If you call:
 - If the actual count of that face value among all dice is less
than the bid, the last bidder loses one die.
 - Otherwise, the caller loses one die.
A player who reaches 0 dice is eliminated. The last
remaining player wins.
[GAME]
New round - Remaining dice:   Player 0: 5;   Player 1: 5
Your current Dice arre: 1, 6, 5, 1, 1
```

```
[GAME] You are Player 0 in Connect Four.
Your disc symbol: X.
The game board has 6 rows and 7 columns.
Players take turns dropping their disc into one of the columns
(0 to 6).
The first to connect (their own) four discs vertically,
horizontally, or diagonally wins.
On your turn, enter the column number in squared brackets
to make your move.
For example: '[col 4]' or '[col 1]'.
[GAME] Board state:
0 1 2 3 4 5 6
-------------
. . . . . . .
. . . . . . .
. . . . . . .
. . . . . . .
. . . . . . .
. . . . . . .
```

Liars Dice — Connect Four

Figure 8: Example observations of five evaluation game environments.

| Parameter | Value |
|---|---|
| ACTOR | |
| Maximum response length | 8192 tokens |
| Sampling temperature | 1.0 |
| (top P, top k) | (1.0, -1) |
| LEARNER | |
| Optimizer | AdamW |
| Adam parameters $(\beta_1, \beta_2)$ | (0.9, 0.95) |
| Weight decay | 0.0 |
| Gradient norm clipping | 1.0 |
| Batch size | 128 |
| Discount factor | 1.0 |
| EMA decay rate | 0.95 |
| Learning rate scheduler | Constant |
| Learning rate | $1 \times 10^{-6}$ |
| Inner proximal update epoch | 2 |
| KL loss coefficient | 0.0 |
| KL penalty coefficient | 0.0 |
| Policy clipping parameter | 0.2 |

Table 6: Hyperparameter configurations used in all experiments.

## D.3 EVALUATION SETTINGS

To investigate whether the reasoning abilities developed through gameplay could transfer to non-game contexts, we evaluate our models on a suite of established benchmarks. All evaluations on these benchmarks are conducted in a zero-shot setting[4] to determine if game-induced reasoning could be successfully transferred to general problem-solving. We use a sampling temperature of $0.6$ and top-p of $0.95$ for all evaluations.

---

[4]Except for the base model, for which we provide few-shot examples that follow the Qwen3 Report settings.

**Math Benchmarks.** For mathematical reasoning, we use MATH500 (Hendrycks et al., 2021), OlympiadBench (He et al., 2024), Minerva Math (Lewkowycz et al., 2022), AIME24, AIME25 (MAA, a), and AMC23 (MAA, b) datasets, which cover a wide range of topics including algebra, geometry, and competitive mathematics. Following the settings in Zhou et al. (2025), we report AVG@32 for AIME24, AIME25 and AMC23; and PASS@1 for other math benchmarks.

**General Reasoning Benchmarks.** For general reasoning, we utilize GPQA-Diamond (Rein et al., 2024), which consists of graduate-level science questions, and MMLU-Pro (Wang et al., 2024), a benchmark for multidisciplinary knowledge.

# E    ADDITIONAL RESULTS AND ANALYSIS

## E.1    DETAILED EVOLUTION OF CASE-BY-CASE ANALYSIS

To understand how reasoning patterns develop during training, we tracked the evolution of case-by-case analysis across checkpoints. Table 7 shows a concrete example from Minerva Math Problem 135, illustrating how models progressively develop structured reasoning.

This progression demonstrates how competitive self-play forces models to develop increasingly structured approaches. Early attempts show unorganized reasoning, while later checkpoints exhibit clear case separation and systematic analysis, a pattern that emerges from game playing and transfers to mathematical problem solving.

## E.2    EXTENDED ANALYSIS OF RAE ABLATION STUDY

Our ablation study of Role-conditioned Advantage Estimation reveals additional insights beyond those presented in Section 4 Figure 9 shows three additional metrics that further demonstrate RAE's critical role in stable self-play training.

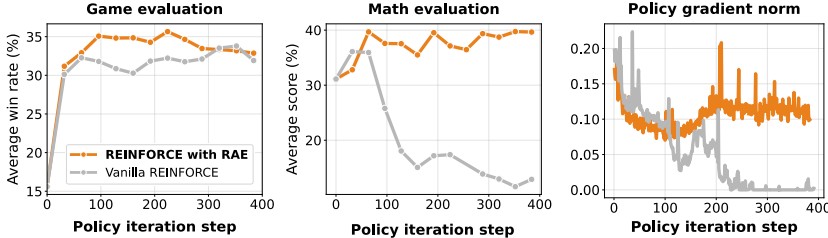

Figure 9: Extended training dynamics comparing REINFORCE with RAE versus vanilla REIN-FORCE (continued from Figure 6). **Left**: Game win rates showing RAE achieves faster initial learning compared to vanilla REINFORCE. **Middle**: Math reasoning performance crashes from 35% to 12% without RAE. **Right**: Policy gradient norms exhibit instability then collapse to near-zero without RAE, while RAE maintains stable gradients around 0.1.

**Game Performance.** The left panel shows that REINFORCE with RAE learns significantly faster, rapidly reaching 35% win rate while vanilla REINFORCE exhibits slower initial learning. RAE's superior learning efficiency demonstrates the benefits of role-conditioned advantage estimation for accelerating policy improvement.

**Math Reasoning Collapse.** The middle panel reveals the most dramatic failure: math reasoning performance without RAE crashes from 35% to 12% at around step 150 (a 66% relative decrease). This collapse occurs precisely when models stop generating reasoning traces, confirming that thinking collapse directly causes reasoning failure.

**Gradient Stability.** The right panel shows policy gradient norms, revealing the underlying optimization dynamics. Without RAE, gradients exhibit high variance with erratic spikes before collapsing to near-zero after step 200, indicating convergence to a degenerate policy. RAE maintains stable gradient norms around 0.1 throughout training, enabling continuous improvement.

Table 7: Case-by-Case Analysis in Minerva Problem 135

| Stage | Problem: Round cosmic ages to nearest power of 10 |
|---|---|
| **Early**(Ckpt 16) | **Approach:** *Lumped reasoning without explicit case separation*
<output> The age when helium formed is approximately 10 minutes... The age when hydrogen formed is 377,000 years... The current age is 13.8 Gyr. Rounding to nearest power of 10, we get: $\boxed{14}$ Gyr </output>
**Case Structure:** No explicit cases - all subproblems handled in continuous text
**Result:** Subproblem 2: 14 Gyr ✗ (should be 10) |
| **Mid**(Ckpt 128) | **Approach:** *Explicit enumeration of cases with structured analysis*
<output>
**Case 1 - Subproblem 0:**
1. Understanding the Context: BBN at $\sim$20 minutes
2. Nearest Power of 10: $10^1 = 10$
3. Final Answer: $\boxed{1}$ minute
**Case 2 - Subproblem 1:**
1. Understanding the Context: Recombination at $\sim$377,000 years
2. Nearest Power of 10: $10^5 = 100,000$
3. Final Answer: $\boxed{400000}$ years
**Case 3 - Subproblem 2:**
1. Understanding the Context: Current age $\sim$13.8 Gyr
2. Nearest Power of 10: Identifies $10^{10}$ correctly
3. Final Answer: $\boxed{13.8}$ Gyr </output>
**Case Structure:** Clear case separation with consistent 3-step process
**Result:** Still incorrect final rounding ✗ |
| **Late**(Ckpt 400) | **Approach:** *Systematic case-by-case with correct execution*
<output>
**Case 1 - Subproblem 0:**
1. Understand Context: He formation in early universe
2. Estimate Time: $\sim$20 minutes after Big Bang
3. Round to Power of 10: $\boxed{20}$ minutes
**Case 2 - Subproblem 1:**
1. Understand Context: H atom formation (recombination)
2. Estimate Time: $\sim$380,000 years
3. Round to Power of 10: $\boxed{400000}$ years
**Case 3 - Subproblem 2:**
1. Understand Context: Current universe age
2. Convert Units: 13.8 billion years = 13.8 Gyr
3. Round to Power of 10: $13.8 \rightarrow 10^1 = \boxed{10}$ Gyr </output>
**Case Structure:** Complete systematic enumeration with correct logic
**Result:** All cases solved correctly ✓ |

These additional results reinforce our main finding: self-play alone is insufficient for stable reasoning development. Proper variance reduction through role-specific baselines is essential to prevent models from converging to degenerate policies that abandon reasoning in favor of minimal outputs.

### E.3 GAME GENERALIZATION

We conducted additional experiments on higher-complexity variants of our training games (5x5 Tic-Tac-Toe, 5-Card Kuhn Poker, and 8-Resource Negotiation) to assess scalability.

Table 8: Generalization performance on increased complexity environments. SPIRAL retains significantly higher performance on out-of-distribution (OOD) tasks compared to SFT.

| Environment | Setting | Qwen3-4B Base | SFT | SPIRAL (Ours) |
|---|---|---|---|---|
| **Tic-Tac-Toe** | 3×3 (Train) | 17.5 | 46.9 | 54.3 |
| | 5×5 (OOD) | 5.5 | 17.8 | 27.3 |
| **Kuhn Poker** | 3-Card (Train) | 21.5 | 43.6 | 53.9 |
| | 5-Card (OOD) | 21.9 | 28.6 | 50.1 |
| **Simple Negotiation** | 5 Resources (Train) | 15.6 | 26.7 | 33.2 |
| | 8 Resources (OOD) | 5.5 | 8.7 | 31.0 |
| **Average (Train)** | | 18.2 | 39.1 | **47.1** |
| **Average (OOD)** | | 11.0 | 18.4 | **36.1** |

Table 8 shows that although SFT has comparable improvement over the base model in in-domain environments (averaging $39.1\%$ vs. $18.2\%$), it struggles to adapt to OOD cases with increased complexity. The SFT model's performance drops to only $18.4\%$. In contrast, SPIRAL demonstrates better generalizability. It not only outperforms SFT in the training settings (averaging $47.1\%$) but maintains robust performance in the more complex OOD environments, achieving an average of $36.1\%$. This suggests that SPIRAL is more capable of handling increased complexity.

### E.4 COMPREHENSIVE BENCHMARK RESULTS

Table 9 presents extended results showing SPIRAL's performance across different training configurations and base models.

These results reveal several important insights. First, single-game SPIRAL training (40.0-41.4% average) outperforms supervised fine-tuning on 25,000 expert examples (38.4% average), validating that self-play can discover more effective reasoning strategies than imitating expert demonstrations. Second, multi-game training (42.3-42.7% average) consistently outperforms single-game variants, suggesting that diverse cognitive challenges create more robust reasoning capabilities. Third, SPIRAL improves even strong models like DeepSeek-Distill-Qwen-7B (from 59.7% to 61.7%), demonstrating that competitive game self-play training can enhance models that already excel at reasoning tasks.

We curated an additional 27k dataset with Qwen3-32B self-play and evaluated SFT on a total of 52k trajectories but training with 1 epoch. As shown in the table, doubling the SFT data yields no major improvement (e.g., Qwen3-4B Average 39.7% vs 39.7%), while SPIRAL consistently outperforms both SFT baselines. This confirms that the benefits of SPIRAL stem from the reinforcement learning dynamic rather than simply dataset size.

### E.5 STATISTICAL ROBUSTNESS

To address concerns regarding statistical significance, we re-ran our main experiments with **3 random seeds** (seeds 14, 42, 100). As shown in Table 10, SPIRAL consistently outperforms the SFT baseline with narrow confidence intervals, confirming the robustness of our gains.

Table 9: SPIRAL training improves reasoning benchmarks for different base models. We include an additional baseline of SFT on 52k trajectories to demonstrate that simply scaling supervised data does not match the gains from self-play.

| Model | MATH500 | AIME'24 | AIME'25 | OlympiadBench | AMC-23 | Minerva Math | GPQA-D | MMLU-Pro | Average |
|---|---|---|---|---|---|---|---|---|---|
| *Qwen3-4B-Base Family* | | | | | | | | | |
| Qwen3-4B-Base | 73.4 | 9.6 | 6.2 | 33.3 | 42.4 | 29.4 | 30.6 | 47.2 | 34.0 |
| + SFT (Kuhn) | 74.0 | 11.0 | 10.4 | 36.7 | 48.6 | 36.8 | 33.0 | 48.8 | 37.4 |
| + SFT (Multi, 25k) | 74.2 | 13.7 | 11.7 | 37.6 | 51.1 | 40.1 | 37.8 | 51.3 | 39.7 |
| + SFT (Multi, 52k) | 73.4 | 16.2 | 11.9 | 39.9 | 48.7 | 38.7 | 38.2 | 51.2 | 39.7 |
| + Mistral Opponent (KuhnPoker) | 64.0 | 4.3 | 2.1 | 29.8 | 31.6 | 26.1 | 35.6 | 43.6 | 29.6 |
| + Gemini Opponent (KuhnPoker) | 69.2 | 5.2 | 4.7 | 33.8 | 29.8 | 33.8 | 35.3 | 55.5 | 33.4 |
| + SPIRAL (TicTacToe) | 75.6 | 10.0 | 13.3 | 38.5 | 55.0 | **42.6** | 37.6 | 57.7 | 41.3 |
| + SPIRAL (KuhnPoker) | 76.4 | 18.2 | **15.6** | 38.4 | 61.2 | 42.4 | 37.0 | 57.7 | 43.4 |
| + SPIRAL (Negotiation) | 75.6 | 11.7 | 10.2 | 38.1 | 51.7 | 39.3 | 36.7 | 57.0 | 40.0 |
| + SPIRAL (TicTacToe+KuhnPoker) | 76.2 | 11.4 | 10.7 | 40.7 | 57.2 | 41.5 | 35.7 | 57.2 | 41.3 |
| + SPIRAL (Multi-Game) | **78.2** | **19.7** | 13.3 | **41.8** | **61.6** | **42.6** | **40.1** | **58.5** | **44.5** |
| *Qwen3-8B-Base Family* | | | | | | | | | |
| Qwen3-8B-Base | 77.0 | 12.1 | 11.2 | 33.5 | 50.6 | 38.2 | 38.0 | 55.7 | 39.5 |
| + SFT (Multi, 25k) | 82.8 | 19.9 | 15.6 | 45.9 | 63.5 | 40.8 | 41.6 | 58.8 | 46.1 |
| + SFT (Multi, 52k) | 81.8 | 22.6 | 14.0 | 48.8 | 61.6 | 40.0 | 43.1 | 57.4 | 46.2 |
| + SPIRAL (Multi-Game) | **86.6** | **26.2** | **16.8** | **49.6** | **65.2** | **46.3** | **44.6** | **61.1** | **49.6** |
| *Octothinker-8B-Base Family* | | | | | | | | | |
| Octothinker-8B-Base | 65.6 | 1.7 | 0.5 | 26.6 | 33.5 | 25.7 | 22.1 | 30.8 | 25.8 |
| + SFT (Multi, 25k) | 66.0 | 3.3 | 3.8 | 23.9 | 31.0 | 23.8 | 24.9 | 39.1 | 27.0 |
| + SFT (Multi, 52k) | 66.4 | 4.2 | 3.6 | 21.9 | 31.6 | 24.6 | 24.9 | 40.8 | 27.3 |
| + SPIRAL (Multi-Game) | **68.6** | **5.3** | **4.8** | **33.7** | **43.2** | **32.0** | **33.8** | **49.3** | **33.8** |
| *Llama-3.1-8B-Instruct Family* | | | | | | | | | |
| Llama-3.1-8B-Instruct | 46.4 | 4.6 | 0.7 | 13.8 | 23.3 | 22.8 | 30.2 | 49.1 | 23.9 |
| + SFT (Multi, 25k) | **51.8** | 4.6 | 0.7 | 19.1 | 23.3 | 21.7 | 30.0 | 48.9 | 25.0 |
| + SFT (Multi, 52k) | 51.0 | 4.3 | 0.0 | **21.5** | 23.2 | 21.1 | 30.2 | 49.1 | 25.1 |
| + SPIRAL (Multi-Game) | 49.8 | **4.9** | **1.8** | 17.3 | **26.0** | **24.6** | **32.2** | **50.4** | **25.9** |
| *DeepSeek-Distill-Qwen-7B Family* | | | | | | | | | |
| DeepSeek-Distill-Qwen-7B | 90.8 | 53.0 | 39.5 | 56.9 | **89.3** | 48.2 | 48.6 | 57.1 | 60.4 |
| + SFT (Multi) | 91.8 | 49.3 | 36.6 | 52.4 | 88.2 | 48.2 | 44.5 | 55.6 | 58.3 |
| + SPIRAL (Multi-Game) | **93.0** | **54.1** | **40.8** | **57.9** | **89.3** | **51.1** | **49.6** | **58.9** | **61.8** |

Table 10: Comprehensive performance comparison across multiple seeds. SPIRAL demonstrates consistent improvement over baselines across diverse benchmarks.

| Model | Math500 | AIME24 | AIME25 | Olympiad | AMC-23 | Minerva | GPQA-D | MMLU-Pro | Average |
|---|---|---|---|---|---|---|---|---|---|
| **Qwen3-4B-Base** | 73.4 | 9.6 | 6.2 | 33.3 | 42.4 | 29.4 | 30.6 | 47.2 | 34.0 |
| + SFT-Multi | $74.0 \pm 0.6$ | $12.4 \pm 1.5$ | $11.2 \pm 1.1$ | $37.8 \pm 0.3$ | $52.2 \pm 0.8$ | $40.7 \pm 0.5$ | $37.7 \pm 1.2$ | $50.9 \pm 0.5$ | $39.6 \pm 0.4$ |
| + SPIRAL-Multi (Ours) | $\mathbf{78.7 \pm 2.0}$ | $\mathbf{18.8 \pm 2.5}$ | $\mathbf{15.0 \pm 1.3}$ | $\mathbf{41.8 \pm 1.3}$ | $\mathbf{62.0 \pm 1.6}$ | $\mathbf{42.1 \pm 1.3}$ | $\mathbf{39.1 \pm 3.1}$ | $\mathbf{58.4 \pm 0.5}$ | $\mathbf{44.5 \pm 0.5}$ |

## E.6 GAME TRAJECTORY STATISTICS

In Table 11, we added a comprehensive table with the average number of game lengths, reasoning tokens per step, P1/P2 self-play win rates, and average win-rate and win-rate per game against Gemini-2.0-Flash.

Table 11: Game trajectory statistics across training checkpoints. We observe increasing game length and reasoning tokens alongside improved win rates against the fixed opponent.

| Training Steps | Avg. Game Round (Moves) | Avg. Reasoning Tokens / Step | Self-Play P1 Win-Rate | Self-Play P2 Win-Rate | Avg. Win-Rate vs. Gemini | Win-Rate vs. Gemini (TicTacToe) | Win-Rate vs. Gemini (Kuhn Poker) | Win-Rate vs. Gemini (Simple Negotiation) |
|---|---|---|---|---|---|---|---|---|
| **Step 0** | 1.69 | 4061 | 42.7% | 57.3% | 12.5% | 12.5% | 6.25% | 18.8% |
| **Step 128** | 7.63 | 1609 | 48.7% | 51.3% | 28.5% | 16.7% | 37.5% | 31.2% |
| **Step 256** | 8.47 | 1755 | 53.2% | 46.8% | 51.8% | 42.9% | 68.8% | 43.8% |
| **Step 384** | 9.43 | 1921 | 59.1% | 40.9% | 66.7% | 75.0% | 68.8% | 56.3% |
| **Step 400** | 9.55 | 2032 | 62.4% | 37.6% | 67.4% | 83.3% | 62.5% | 56.3% |

## E.7 SPIRAL AS PART OF THE MID-TRAINING STAGE

We're running Base→SPIRAL→RLVR experiments to compare with Base→RLVR in terms of convergence speed and final performance. Specifically, we use Math-12k (Lightman et al., 2023) for the standard Math RLVR training, and we examine the two variants: Base→SPIRAL→RLVR and Base→RLVR→SPIRAL.

Our results in Table 12 validate SPIRAL's effectiveness on integrating as part of the mid-training: SPIRAL → RLVR outperforms the Base → RLVR baseline, while applying SPIRAL as a post-RLVR stage (RLVR → SPIRAL) yields the highest performance (Avg 48.1). Furthermore, on the

Table 12: Performance comparison of SPIRAL integrated into different training stages. SPIRAL acts as a robust performance booster both before and after RLVR.

| Model | Math500 | AIME24 | AIME25 | Olympiad | AMC-23 | Minerva | GPQA-D | MMLU-Pro | Average |
|---|---|---|---|---|---|---|---|---|---|
| **Qwen3-4B-Base** | 73.4 | 9.6 | 6.2 | 33.3 | 42.4 | 29.4 | 30.6 | 47.2 | 34.0 |
| RLVR (Math) | 83.0 | 18.4 | 15.6 | 44.6 | 62.8 | 43.4 | 43.7 | 56.8 | 46.0 |
| **SPIRAL** | 78.2 | 19.7 | 13.3 | 41.8 | 61.6 | 42.6 | 40.1 | 58.5 | 44.5 |
| **SPIRAL → RLVR** | 84.2 | **23.1** | 17.2 | 45.2 | 59.6 | 42.1 | 43.2 | 57.4 | 46.5 |
| **RLVR → SPIRAL** | **86.1** | 22.6 | **18.1** | **46.0** | 62.5 | **44.3** | **44.7** | **60.8** | **47.9** |

stronger DeepSeek-Distill-Qwen-7B, SPIRAL-Multi improves the average score to 59.3, whereas standard SFT leads to performance regression. These findings demonstrate that SPIRAL serves as a robust booster that mitigates the alignment tax often seen in standard fine-tuning stages.

### E.8 INSTRUCT MODEL RESULTS

To better study our method's effectiveness, we perform additional experiments with **Qwen3-4B-Instruct-2507**.

As shown in the Table 13, standard SFT on reasoning trajectories slightly degrades performance. In contrast, **SPIRAL** successfully reverses this trend. SPIRAL achieves a **2% improvement** over the original model and a substantial average gain of **3.6%** over the SFT baseline, reaching a total average score of **75.51%**. This demonstrates that self-play can refine genuine reasoning capabilities even in models that have strong performance.

Table 13: Performance comparison on Qwen3-4B-Instruct. While standard SFT leads to regression, SPIRAL improves reasoning capabilities across most benchmarks.

| Model | Math500 | AIME24 | AIME25 | Olympiad | AMC-23 | Minerva | GPQA-D | MMLU-Pro | Average |
|---|---|---|---|---|---|---|---|---|---|
| **Qwen3-4B-Instruct** (Base) | 91.2 | 64.6 | 47.4 | **89.6** | 82.1 | **87.6** | **62.0** | 69.6 | **74.10** |
| + SFT-Multi | 90.8 | 59.3 | 46.1 | 86.4 | 79.6 | 82.3 | 61.4 | 69.1 | **71.88** |
| + SPIRAL-Multi (Ours) | **93.5** | **67.9** | **49.8** | 88.8 | **83.3** | 85.5 | 61.8 | **71.5** | **75.91** |

## F CASE STUDY METHODOLOGY

This section details our systematic approach to discovering and analyzing reasoning pattern transfer from games to mathematics. Rather than searching for predetermined patterns, we employed a bottom-up discovery process to identify what reasoning strategies naturally emerge and transfer between domains.

### F.1 DATA COLLECTION FRAMEWORK

Our analysis examined reasoning traces from two sources across three training checkpoints:

**Game Trajectories:** We collected 290 complete Kuhn Poker games, focusing on winning trajectories to identify successful reasoning strategies. Each trajectory includes the complete thought process from initial card observation through final decision.

**Mathematical Solutions:** We analyzed 46,792 solution attempts across MATH500, AIME, OlympiadBench, and Minerva Math benchmarks. Solutions were categorized by success (score=1) or failure (score=0) to understand which reasoning approaches prove effective.

**Temporal Analysis:** Checkpoints at steps 0 (initial), 128 (intermediate), and 400 (final) capture the evolution of reasoning complexity throughout training.

### F.2 BOTTOM-UP PATTERN DISCOVERY PROCESS

Rather than searching for predefined patterns, we employed GPT-4.1 to discover patterns that naturally emerge in the data. This bottom-up approach ensures we capture the actual reasoning strategies used rather than imposing our expectations.

---

**Pattern Discovery Prompt**

```
Analyze these {domain} reasoning traces using a BOTTOM-UP approach.
Don't look for predefined patterns.  Instead, discover what
reasoning patterns actually exist.
REASONING TRACES ({len(sample)} samples from {len(traces)} total):
{json.dumps([t['reasoning'] for t in sample[:40]], indent=2)}
Your task:
     1. Read through ALL the reasoning traces carefully
     2. Identify RECURRING patterns or structures that appear
        multiple times
     3. Group similar reasoning approaches together
     4. Name each discovered pattern based on what it actually does
     5. Count how many traces use each pattern
     6. Provide example quotes for each pattern
Format your response as:
PATTERN 1:  [Descriptive Name]
- Description:  [What this pattern does]
- Count:  X/{len(sample)} traces
- Example quotes:  [2-3 actual quotes showing this pattern]
Be specific and grounded in the actual data.  If you see a pattern
only once, don't include it.
```

---

This discovery process revealed three dominant patterns that emerged independently in both domains:

1. **Case-by-Case Analysis:** Systematic enumeration of scenarios
2. **Expected Value Calculation:** Probabilistic decision-making
3. **Pattern Recognition:** Identifying regularities and structures

### F.3 CROSS-DOMAIN TRANSFER QUANTIFICATION

After discovering patterns in each domain, we compared them to identify which strategies transfer between games and mathematics:

---

**Pattern Comparison Prompt**

```
Compare the reasoning patterns discovered in games vs mathematics:
GAME PATTERNS:
{game_patterns}
MATH PATTERNS:
{math_patterns}
Analyze:
     1. Which patterns appear in BOTH domains?  (These show transfer)
     2. Which patterns are unique to each domain?
     3. Calculate transfer rates for shared patterns
     4. Identify the most successfully transferred reasoning
        strategies
     5. Explain WHY certain patterns transfer well
Focus on concrete evidence of transfer, not speculation.
```

---

The transfer analysis revealed:

**Case-by-Case Analysis** shows near-perfect transfer (72% in games to 71% in math) because systematic enumeration represents domain-agnostic structured thinking. Whether analyzing opponent possibilities in Poker or solution branches in mathematics, the core cognitive skill remains identical.

**Expected Value Calculation** exhibits limited transfer (78% in games to 28% in math) because explicit probabilistic decision-making appears primarily in probability and optimization problems. Most mathematical domains lack the decision-theoretic structure that makes this pattern universally applicable in games.

**Pattern Recognition** demonstrates amplification during transfer (35% in games to 45% in math). Mathematics inherently requires pattern identification, so game training enhances an already-essential mathematical skill, producing stronger pattern recognition than games alone develop.

### F.4 PATTERN EVOLUTION ANALYSIS

To understand how reasoning develops during training, we tracked pattern emergence across checkpoints:

```
Evolution Analysis Prompt

Analyze how reasoning patterns evolve across training checkpoints:
{json.dumps(evolution_analysis, indent=2)}
For each checkpoint:
    1. Describe the complexity of reasoning
    2. Identify new patterns that emerge
    3. Track how patterns become more sophisticated
    4. Show concrete examples of improvement
Focus on the actual evolution you can see in the data.
```

### F.5 CONCRETE TRANSFER EXAMPLE IDENTIFICATION

To validate transfer claims, we identified parallel reasoning structures across domains:

```
Transfer Example Prompt

Find the clearest examples of reasoning transfer from games to
mathematics:
{json.dumps(examples, indent=2)}
For each complexity level (short/medium/long):
    1. Identify parallel reasoning structures
    2. Quote specific passages that show transfer
    3. Explain what cognitive skill is being transferred
    4. Rate the clarity of transfer (1-10)
Focus on examples where the same reasoning approach clearly appears
in both domains.
```

### F.6 PATTERN CLASSIFICATION AT SCALE

After discovering patterns through bottom-up analysis, we classified all traces to measure transfer rates:

---

**Pattern Classification Prompt**

```
Analyze these {domain} reasoning traces and find examples of each
pattern.
PATTERNS TO FIND:
    1. Case-by-Case Analysis:  Systematic enumeration of different
       scenarios/cases

    2. Expected Value Calculation:  Explicit probability
       calculations, computing expected outcomes

    3. Pattern Recognition:  Identifying recurring structures,
       noticing trends
REASONING TRACES:
{json.dumps(batch, indent=2)}
For EACH pattern, identify ALL examples that clearly demonstrate
it.
Return in this EXACT format:
CASE_BY_CASE_INDICES: [list of indices]
EXPECTED_VALUE_INDICES: [list of indices]
PATTERN_RECOGNITION_INDICES: [list of indices]
Be strict - only include clear examples of each pattern.
```

### F.7 VALIDATION METHODOLOGY

To ensure robust findings, we implemented multiple validation steps:

**Sampling Strategy:** We analyzed 50 random trajectory samples per checkpoint to avoid selection bias while maintaining computational feasibility.

**Success Stratification:** Separate analysis of successful and failed attempts revealed which reasoning strategies genuinely contribute to problem-solving rather than merely appearing frequently.

**Manual Verification:** Spot-checking GPT-4.1's pattern classifications against raw traces confirmed the accuracy of automated analysis.

**Scale Validation:** After discovering patterns through focused analysis, we classified all 46,792 mathematical traces to verify that observed transfer rates hold at scale.

This methodology ensures our findings reflect genuine cognitive transfer rather than superficial pattern matching, providing quantitative evidence that competitive gameplay develops reasoning skills applicable far beyond the training domain.

## G  GAME ENVIRONMENT SPECIFICATIONS

This section provides detailed specifications for all game environments used in our experiments, including both training and evaluation games.

### G.1 TRAINING GAME ENVIRONMENTS

**TicTacToe** tests spatial pattern recognition through perfect information gameplay. Players alternate placing marks on a 3×3 grid, aiming to create lines of three. The deterministic nature isolates pure strategic reasoning from uncertainty management. Success requires recognizing winning patterns, blocking opponent threats, and creating fork positions that guarantee victory. We hypothesize these skills transfer to geometric reasoning and spatial visualization tasks.

**Kuhn Poker** introduces probabilistic reasoning through minimal hidden information. With only three cards (Jack, Queen, King), one per player plus one undealt, the game distills poker to essential elements of bluffing and value betting. Players can check, bet, call, or fold, with outcomes determined by card strength. Success requires calculating expected values, modeling opponent behavior, and making decisions under uncertainty. These capabilities should transfer to probability problems and strategic decision-making.

**Simple Negotiation** is a game that develops multi-constraint optimization skills through resource trading. Two players exchange Wheat, Wood, Sheep, Brick, and Gold tokens. Since the utility of these resources varies for each player, there is a natural incentive to trade. Each player aims to maximize their portfolio's value by making proposals and counteroffers. Success requires understanding an opponent's preferences, planning multi-step trades, and communicating strategically. We expect these skills to improve performance on optimization problems and multi-constraint reasoning tasks.

### G.2 OUT-OF-DISTRIBUTION EVALUATION GAMES

Our evaluation suite tests whether learned skills generalize to novel mechanics:

**Snake** extends spatial reasoning to dynamic environments. Players control snakes navigating grids to collect apples while avoiding collisions with walls, themselves, or opponents. This tests whether static pattern recognition from TicTacToe transfers to trajectory planning and dynamic obstacle avoidance.

**Pig Dice** isolates risk-reward decision making. Players repeatedly roll dice to accumulate points but lose all turn points when rolling 1. This tests whether probabilistic reasoning from Kuhn Poker extends to sequential risk assessment and expected value calculation in different contexts.

**Truth and Deception** focuses on asymmetric information and persuasion. One player knows the true fact among options and misleads through conversation while the other must identify truth through questioning. This evaluates whether negotiation skills transfer to pure communication strategy.

These diverse evaluation games probe different aspects of transfer learning, revealing which cognitive skills generalize beyond their training context and confirming that SPIRAL develops fundamental reasoning capabilities rather than game-specific tactics.

