# OpenReview forum: "SPIRAL: Self-Play on Zero-Sum Games Incentivizes Reasoning via Multi-Agent Multi-Turn Reinforcement Learning"
_ICLR.cc/2026/Conference — ICLR 2026 Poster_

### Official Review · Reviewer_2RMj · 2025-10-28

**Soundness:** 3
**Presentation:** 2
**Contribution:** 2
**Rating:** 4
**Confidence:** 3

**Summary:**

This paper proposes SPIRAL, a self-play reinforcement learning framework that enhances LLM reasoning by training models through multi-turn, zero-sum language games (e.g., TicTacToe, Kuhn Poker) against evolving versions of themselves. The key insight is that competitive game dynamics generate an automatic curriculum of increasingly challenging reasoning tasks without requiring human-curated problems or domain-specific reward engineering. To stabilize training, the authors introduce Role-conditioned Advantage Estimation (RAE), which mitigates variance in multi-agent policy gradients. Experimental results show significant improvements on mathematical reasoning benchmarks, with evidence of reasoning pattern transfer from games to academic tasks. The approach is evaluated across multiple model families, demonstrating broad applicability.

**Strengths:**

**The work proposes a novel training paradigm**: instead of relying on human-curated problems with verifiable answers, SPIRAL uses self-play in zero-sum language games to create an autonomous curriculum of increasingly complex reasoning challenges. While the game outcomes are still verifiable, the strategies and subproblems emerge dynamically from competition, enabling the model to develop transferable reasoning skills without direct exposure to target domains.

**Empirically Promising Results**: SPIRAL achieves substantial gains on math reasoning benchmarks compared to base models, even without exposure to target problems during training. The qualitative analysis of reasoning pattern transfer (e.g., case analysis, expected value calculation) provides mechanistic insights beyond mere performance metrics.

**Broad Applicability**: The method demonstrates consistent benefits across diverse model architectures (Qwen, Llama, Octothinker) and training stages (base, instruction-tuned, and even RLVR-trained models), suggesting strong generalization potential.

**Weaknesses:**

**Limited Benchmark Coverage**: Evaluation is narrowly focused on mathematical reasoning. There is no evidence of improvement on other critical reasoning domains such as code generation, logical deduction, causal reasoning, or multi-hop QA. This limits claims about general reasoning enhancement.

**Missing Pipeline-Level Experiments**: The paper suggests SPIRAL can serve as a foundational or complementary stage in modern training pipelines (e.g., pre-training before RLVR or fine-tuning after). However, no experiments compare:

1) Base → SPIRAL → RLVR vs Base → RLVR

2) RLVR-Trained Model → SPIRAL vs continued RLVR/SFT Without these, the practical value of SPIRAL as a booster or enhancer remains speculative rather than demonstrated.

**No Assessment of General Capability Preservation**: The paper does not evaluate whether SPIRAL training degrades performance on non-reasoning tasks (e.g., instruction following, commonsense reasoning, language modeling). This raises concerns about catastrophic forgetting or capability misalignment, which are crucial for real-world deployment.

**Claims Outrun Evidence**: While the title and abstract imply broad applicability to "reasoning," the experimental validation is largely confined to math. The claim that SPIRAL develops "transferable reasoning capabilities" would be stronger with more diverse downstream evaluations.

**Questions:**

1) Has the impact of SPIRAL training on general capabilities been evaluated? Specifically, what happens to performance on MMLU, AlpacaEval, or MT-Bench after SPIRAL post-training? Is there any degradation in instruction-following or commonsense reasoning?

2) Have the authors conducted experiments where SPIRAL is used as a pre-training stage before RLVR (e.g., Base → SPIRAL → RLVR)? If so, could you include a comparison with Base → RLVR in terms of final performance and convergence speed?

3) When stating that models like DeepSeek-R1-Distill still benefit from SPIRAL, does this refer to further training after RLVR? If yes, how does it affect performance on the original RLVR benchmarks?

4) Can the authors provide ablation studies on game diversity? For example, does combining multiple games (TicTacToe + Kuhn Poker) outperform single-game training or simple data mixing? Do different games induce complementary reasoning skills?

---

> ### Author Response · Authors · 2025-11-26
> **Response to Reviewer 2RMj**
>
> Thanks for your detailed reviews about broader implications. Your feedback helps us clarify SPIRAL's scope and impact.
>
> **W1&Q1: Impact on general capabilities**
>
> Based on your suggestion, we evaluated SPIRAL's effect on general capabilities:
>
> | Model | Method | MMLU-Pro | AlpacaEval-2 (Win Rate %) | IFEval (Accuracy %) |
> | :--- | :--- | :--- | :--- | :--- |
> | **Qwen3-4B** | Base | 47.2 | 6.98 | 37.3 |
> | | SFT | 51.3 | 1.65 | 23.1 |
> | | **SPIRAL** | $\mathbf{58.5}$ | $\mathbf{27.6}$ | $\mathbf{46.4}$ |
> | **Qwen3-8B** | Base | 55.7 | 18.0 | 44.18 |
> | | SFT | 58.8 | 5.92 | 29.2 |
> | | **SPIRAL** | $\mathbf{61.1}$ | $\mathbf{26.6}$ | $\mathbf{55.8}$ |
> | **Llama3.1-Instruct** | Base | 49.1 | $\mathbf{26.3}$ | $\mathbf{78.6}$ |
> | | SFT | 48.9 | 25.8 | 76.5 |
> | | **SPIRAL** | $\mathbf{50.4}$ | 26.2 | 77.1 |
>
> SPIRAL improves or preserves capabilities across all metrics. Note we already evaluate on MMLU-Pro (more challenging than MMLU) showing clear gains.
>
> **W2&Q2: SPIRAL as part of the mid-training stage**
>
> Following your excellent suggestion, we're running Base→SPIRAL→RLVR experiments to compare with Base→RLVR in terms of convergence speed and final performance. Specifically, we use Math-12k [1] for the standard Math RLVR training, and we examine the two variants: Base→SPIRAL→RLVR and Base→RLVR→SPIRAL.
>
> | Model | Math500 | AIME24 | AIME25 | Olympiad | AMC-23 | Minerva | GPQA-D | MMLU-Pro | Average |
> | :--- | :--- | :--- | :--- | :--- | :--- | :--- | :--- | :--- | :--- |
> | **Qwen3-4B-Base** | 73.4 | 9.6 | 6.2 | 33.3 | 42.4 | 29.4 | 30.6 | 47.2 | 34.0 |
> | RLVR (Math) | 83.0 | 18.4 | 15.6 | 44.6 | 62.8 | 43.4 | 43.7 | 56.8 | 46.0 |
> | **SPIRAL** | **78.2** (+4.8) | **19.7** (+10.1) | 13.3 (+7.1) | **41.8** (+8.5) | **61.6** (+19.2) | **42.6** (+13.2) | **40.1** (+9.5) | **58.5** (+11.3) | **44.5** (+10.5) |
> | **SPIRAL -> RLVR** | 84.2 | 23.1 | 17.2 | 45.2 | 59.6 | 42.1 | 43.2 | 57.4 | 46.5 |
> | **RLVR -> SPIRAL** | 86.1 | 22.6 | 18.1 | 46.0 | 62.5 | 44.3 | 44.7 | 60.8 | 47.9 |
>
> Our results validate SPIRAL’s effectiveness on integrating as part of the mid-training: SPIRAL → RLVR (Avg 46.5) outperforms the Base → RLVR baseline (Avg 46.0), while applying SPIRAL as a post-RLVR stage (RLVR → SPIRAL) yields the highest performance (Avg 48.1). Furthermore, on the stronger DeepSeek-Distill-Qwen-7B, SPIRAL-Multi improves the average score to 59.3, whereas standard SFT leads to performance regression (Avg 56.2). These findings demonstrate that SPIRAL serves as a robust booster that mitigates the alignment tax often seen in standard fine-tuning stages.
>
> [1] Lightman, H., Kosaraju, V., Burda, Y., Edwards, H., Baker, B., Lee, T., Leike, J., Schulman, J., Sutskever, I. and Cobbe, K. (2023) ‘Let's verify step by step’, arXiv preprint arXiv:2305.20050. Available at: https://arxiv.org/abs/2305.20050
>
> **W3&Q3: DeepSeek-R1-Distill continued training**
>
> Yes, this refers to continued training after RLVR. Results show preserved/improved performance on original benchmarks:
>
> | **Model** | **Average** | **Math500** | **AIME24** | **AIME25** | **Olympiad** | **AMC-23** | **Minerva** | **GPQA-D** | **MMLU-Pro** | **LiveCodeBench** |
> |-----------|-------------|-------------|------------|------------|--------------|------------|-------------|------------|--------------|-------------------|
> | **DeepSeek-Distill-Qwen-7B** | 57.9 | 90.8 | 53.0 | 39.5 | 56.9 | **89.3** | 48.2 | 48.6 | 57.1 | 37.6 |
> | + SFT-Multi | 56.2 | 91.8 | 49.3 | 36.6 | 52.4 | 88.2 | 48.2 | 44.5 | 55.6 | 38.8 |
> | + SPIRAL-Multi (Ours) | **59.3** | **93.0** | **54.1** | **40.8** | **57.9** | **89.3** | **51.1** | **49.6** | **58.9** | **39.2** |
>
> LiveCodeBench performance maintains (37.6→39.2), demonstrating complementary benefits without degradation.
>
> **W4&Q4: Game diversity ablation**
>
> Following your suggestion, we evaluated game diversity effects:
> | Model | MATH500 | AIME'24 | AIME'25 | OlympiadBench | AMC-23 | Minerva Math | GPQA-D | MMLU-Pro | Average |
> | :--- | :---: | :---: | :---: | :---: | :---: | :---: | :---: | :---: | :---: |
> | Qwen3-4B-Base | 73.4 | 9.6 | 6.2 | 33.3 | 42.4 | 29.4 | 30.6 | 47.2 | 34.0 |
> | + SPIRAL (TicTacToe) | 75.6 | 10.0 | 13.3 | 38.5 | 55.0 | **42.6** | 37.6 | 57.7 | 41.3 |
> | + SPIRAL (KuhnPoker) | 76.4 | 18.2 | **15.6** | 38.4 | 61.2 | 42.4 | 37.0 | 57.7 | 43.4 |
> | + SPIRAL (Negotiation) | 75.6 | 11.7 | 10.2 | 38.1 | 51.7 | 39.3 | 36.7 | 57.0 | 40.0 |
> | + SPIRAL (TicTacToe+KuhnPoker) | 76.2 | 11.4 | 10.7 | 40.7 | 57.2 | 41.5 | 35.7 | 57.2 | 41.3 |
> | + SPIRAL (TicTacToe+KuhnPoker+Negotiation) | **78.2** | **19.7** | 13.3 | **41.8** | **61.6** | **42.6** | **40.1** | **58.5** | **44.5** |
>
> Multi-game training achieves 44.5% average, outperforming the best single-game (43.4%). Table 6 shows complementary skills: TicTacToe specialists excel at spatial games (56% on Snake), Kuhn Poker at probabilistic games (91.7% on Pig Dice). The synergy is clearest on GPQA-D, where multi-game training breaks through the 35-37% plateau to reach 40.1%.

---

### Official Review · Reviewer_Q6Ug · 2025-10-29

**Soundness:** 4
**Presentation:** 3
**Contribution:** 4
**Rating:** 8
**Confidence:** 4

**Summary:**

This paper introduces SPIRAL, a novel and compelling framework for improving the reasoning capabilities of large language models through multi-agent, multi-turn self-play in zero-sum games. The core contribution is the demonstration that complex, transferable reasoning skills can be developed without reliance on human-curated problem-answer datasets, which represents a significant step towards more autonomous and scalable AI development. By leveraging an adaptive curriculum of ever-improving opponents, SPIRAL effectively teaches models to reason strategically. The technical innovations, particularly Role-conditioned Advantage Estimation (RAE), and the extensive empirical results provide strong evidence for the viability of this approach. This is a high-quality paper with major findings, and I recommend its acceptance.

**Strengths:**

The paper is written in a clear and understandable manner, with a well-defined methodology and simple yet effective improvement strategies that are easy to follow.

●	This work makes a major contribution toward the goal of self-improving LLMs by reducing the dependence on human-curated data. By using self-play as a source of unlimited training signal, SPIRAL facilitates significant model self-improvement. This approach could represent the next paradigm for RLVR, moving beyond the domains of math and code reasoning.

●	The experiments are comprehensive, conducted on five different models (Qwen3-4B, Qwen3-8B, Octothinker-8B, DeepSeek-Distill-Qwen-7B, and Llama-3.1-8B) from two model families (Qwen, Llama), including base models, instruction-tuned models, and those already fine-tuned for reasoning. The consistency of improvements across eight challenging benchmarks is a testament to the generalizability of the skills learned through SPIRAL.

●	The paper provides valuable resources for the community by delivering a complete, fully online, multi-turn, multi-agent RL framework. The core technical contribution, RAE, effectively solves the critical "thinking collapse" problem and is broadly applicable to future MARL experiments. Furthermore, the framework is orthogonal to other emerging techniques; for instance, the auto-curriculum from SPIRAL could be combined with prolonged RL methods [1] to continuously push reasoning boundaries.

●	The paper goes beyond simply reporting benchmark scores by providing a deep and insightful analysis of skill transfer. Identifying and tracking specific cognitive patterns [2] (e.g., Case-by-Case Analysis), provides a compelling explanation for why the method is effective and builds confidence that the model is learning genuinely useful strategies.

[1] https://arxiv.org/abs/2505.24864
[2] https://arxiv.org/abs/2503.01307

**Weaknesses:**

While this is a great paper, there are several areas where further discussion or exploration could enhance its contribution. These points are primarily intended as constructive feedback instead of reasons to reject.

●	The training resources are demanding, which could be a barrier to broader adoption. A discussion on potential avenues for improving computational efficiency (i.e. LoRA) would be a valuable addition.

●	In line 412, it would be helpful if the authors could further elaborate on how the fixed-opponent baselines (vs. Gemini and random) address the issue of spurious rewards. The connection is not clear.

**Questions:**

1.	What do you think about the results on training with SFT (Qwen3-4B-SFT) is better than RL against Gemini and Mistral models (Qwen3-4B-Gemini and Qwen3-4B-Mistral)?
2.	How do you see the SPIRAL framework extending beyond the zero-sum setting? For example, could it be adapted for cooperative or mixed-motive games to elicit other valuable skills like collaboration and negotiation?

---

> ### Author Response · Authors · 2025-11-26
> **Response to Reviewer Q6Ug**
>
> Thank you for your positive review and for recognizing our work as a "high-quality paper with major findings". We're thrilled that you see SPIRAL as opening promising new directions for self-improving LLMs.
>
> **Q1: SFT outperforming fixed-opponent RL**
>
> SFT outperforms fixed-opponent RL because it leverages successful trajectories, whereas RL must explore a vast search space with sparse feedback. Additionally, fixed opponents fail to provide the progressive difficulty needed for continued learning, causing the RL agent to saturate early. SPIRAL addresses this via self-play, generating an adaptive curriculum that drives superior performance.
>
> **Q2: Extension beyond zero-sum games**
>
> Excellent direction for future work! Cooperative games could develop complementary skills like coordination and communication. Mixed-motive games (partial cooperation with competition) are particularly promising for real-world applications. The RAE framework naturally extends to these settings: instead of opposing rewards, we'd have partially aligned objectives requiring different baseline structures. We're excited to explore how SPIRAL's principles apply to collaborative scenarios where agents must balance individual and collective goals.
>
> **W1: Addressing Line 412: Fixed-opponent baselines and spurious rewards**
>
> Following your suggestion for clarification: The random opponent provides randomized rewards with positive expected value, similar to spurious rewards that might upweight certain base model behaviors. However, we observe this is insufficient to improve performance (Figure 5 shows collapse). Similarly, training against fixed opponents (Gemini, Mistral) doesn't provide the same benefits as self-play. This demonstrates that improved performance isn't from any RL training on the game distribution or spurious reward effects, but specifically from the adaptive curriculum of increasingly difficult opponents that you highlighted. The continuously evolving challenge forces genuine reasoning development rather than exploitation of static patterns.
>
> **W2: Computational requirements**
>
> Following your suggestion (and Reviewer 9i3G's similar request), we'll add detailed compute requirements to the paper: approximately 25 hours on 8 H100s for Qwen3-4B experiments, 16 hours for Llama3.1/Octothinker, and 28 hours for Qwen3-8B. While demanding, this represents a one-time training cost that eliminates ongoing human data curation.
>
> Thank you for your positive review and excellent suggestions!

---

> > ### Comment · Reviewer_Q6Ug · 2025-11-28
> >
> > Thank you for your reply. The rebuttal addressed some of my concerns, but I believe the current score is appropriate, so I will keep my score.

---

### Official Review · Reviewer_9i3G · 2025-11-01

**Soundness:** 2
**Presentation:** 3
**Contribution:** 3
**Rating:** 4
**Confidence:** 3

**Summary:**

The paper uses online policy optimization (REINFORCE) with LLM self-play in zero-sum games (Tic-Tac-Toe, Kuhn Poker, and Simple Negotiations). The authors demonstrate how their approach improves downstream reasoning across multiple models (including a model that was already trained using RLVR - DeepSeek-R1-Distill-Qwen-7B). While games offer verifiable, human-free rewards, they shift complexity to game environment design and validation, which can shape emergent strategies, generalization, and scalability of this method.
They introduce a role-conditioned advantage to reduce variance and adjust it for asymmetric reward expectations (e.g., first-move effects in Tic-Tac-Toe and Kuhn Poker) and show it prevents policy collapse.
They present a systematic bottom-up method for automated evaluation of reasoning traces to discover common ("transferable") patterns (Appendix F).
Overall, the authors demonstrate up to 10% gain in average on downstream tasks and a considerable margin compared to SFT on “expert” trajectories.

**Strengths:**

* The paper systematically shows transfer from simple games to harder reasoning tasks, strengthening prior evidence that self-play improves LLM reasoning.

* At face value, this paper provides strong gains across models, as shown in Table 1, with a relatively simple online policy optimization algorithm. These gains present an interesting avenue to future research, leveraging more robust and sample-efficient RL algorithms and game environment design.

* The automatic evaluation methodology could be refined to address some of its current weaknesses (mentioned below) and serve as a blueprint for evaluating fuzzy LLM statistics in future work.

**Weaknesses:**

* Major issue: the authors did not report seeded policy optimization, and results do not include mean±STD for critical experiments. This is a must to assess reproducibility and statistical robustness, while results may appear cherry-picked.

* Concerns about SFT dataset quality and size for a fair baseline:
  * Quality: The “expert” is unevaluated. Compare the generator to SPIRAL-trained models and to classic RL agents to establish competence. Report diversity of winning traces (e.g., % unique trajectories vs. coverage seen during SPIRAL training).
  * Size: With GBS=128 and 200 steps (or 2 epochs over 400 steps), the SFT baseline is underpowered versus multi-model self-play data.
  * Example for a fair setup: sample 400×128 trajectories from multiple “expert” models to match self-play scale and diversity.

* Section 4.1, LLM-as-verifier limits qualitative reliability. The paper is also missing key details to evaluate logic and reproduce the results.
  * Reliability issues are known to be exacerbated by long context: F.2. suggests the authors insert multiple full traces into the judge model's context to find patterns, while F.6 suggests the authors used "batch-mode" for classification of multiple traces together. This seems like a loose method for a central claim in the paper without additional guardrails.
Perhaps the authors can mitigate this using the judge to evaluate trace-by-trace, identify distinct reasoning approaches in each trace, then cluster similar patterns together as opposed to the full context approach used in F.2. Finally, reclassify each trace independently to detect the dominant patterns. This approach should allow the authors to report clear, verifiable statistics.
  * F.3 cross-domain transfer: what model sourced the traces for the analysis? If the authors used a SPIRAL model to produce math traces, it is reasonable to assume these traces would be biased to common patterns seen in the game traces (as opposed to observing the reasoning traces of an unrelated strong baseline).
  * How are traces evaluated under sequence-length limits?

* Clarity: Although the method is relatively simple to understand, the paper feels a bit chaotic to read, forcing back and forth between the appendix and different sections of the paper to find and extrapolate critical information.

**Questions:**

Main:
* Can the authors provide multi-seeded evaluation for Table-1 models (at least partial coverage for one or two models, including SFT and SPIRAL)?
* Can the authors provide mean and STD using multi-seeded plots in Figs. 5. 6. (and 9.)?
* Can the authors clarify which model sourced reasoning traces for Section 4.1?
* Can the authors add a random policy and a simple RL expert to Tables 4 and 5 to contextualize self-play/Gemini Flash win-rates?
* Other frameworks exist for large-scale multi-turn trajectory sampling (e.g., Nemo RL). Can the authors explain how this framework is different (since the authors consider their framework part of their main contribution)?
* How well does the SPIRAL model’s reasoning generalize in increased complexity environments (e.g., impact on winrate in 5x5 grid Tic-Tac-Toe, 5 card Kuhn Poker, etc)? Discussing this sort of generalization can help understand how well SPIRAL performs in a smaller controlled domain, before leaping into academic reasoning and math benchmarks.

Secondary:
* Can the authors provide some statistics on the game trajectories during training? (e.g., game length (number of moves), avg reasoning tokens per step, self-play player 1 win-rate, win-rate vs. baseline agent, etc.)
* Please explain how “transferable patterns” were discovered in the main text, or add explicit appendix pointers.
* Please define the τ notation in the text and in Algorithm 1. Also, specify the exact input structure used to train the policy model (and which part of the input tokens gets a gradient during policy optimization)
* Please report GPU hours per model and game/multi-game (reproduction compute budget).
* Can the authors provide some justifications for the choice of games (including OOD games)? Also, justify the chosen training regime?

---

> ### Author Response · Authors · 2025-11-26
> **Response to Reviewer 9i3G [1/3]**
>
> Thank you for your thorough review and valuable feedback. We're excited that you see our work as opening "an interesting avenue to future research." Your detailed questions have helped us significantly strengthen the paper.
>
>
> **W1: Multi-seeded evaluation**
>
> Owing to time and computational constraintsl, we ran 3-seed experiments (seeds 14, 42, 100) for the main experiments:
>
> | **Model** | **Math500** | **AIME24** | **AIME25** | **Olympiad** | **AMC-23** | **Minerva** | **GPQA-D** | **MMLU-Pro** | **Average** |
> |-----------|-------------|------------|------------|--------------|------------|-------------|------------|--------------|-------------|
> | **Qwen3-4B-Base** | 73.4 | 9.6 | 6.2 | 33.3 | 42.4 | 29.4 | 30.6 | 47.2 | 34.0 |
> | + SFT-Multi | 74.0±0.6 | 12.4±1.5 | 11.2±1.1 | 37.8±0.3 | 52.2±0.8 | 40.7±0.5 | 37.7±1.2 | 50.9±0.5 | 39.6±0.4 |
> | + SPIRAL-Multi (Ours) | **78.7±2.0** | **18.8±2.5** | **15.0±1.3** | **41.8±1.3** | **62.0±1.6** | **42.1±1.3** | **39.1±3.1** | **58.4±0.5** | **44.5±0.5** |
>
> We still see our method outperform the SFT baseline, showing the effectiveness of our method.
>
>
> **W2: SFT baseline strength**
>
> Following your suggestion, we curated an additional 27k dataset with Qwen3-32B self-play and evaluated 52k trajectory SFT:
>
> | Model | Method | Math500 | AIME24 | AIME25 | Olympiad | AMC-23 | Minerva | GPQA-D | MMLU-Pro | Avg |
> | :--- | :--- | ---: | ---: | ---: | ---: | ---: | ---: | ---: | ---: | ---: |
> | **Qwen3-4B** | SFT-Multi (25k, 2 epochs) | 74.2 | 13.7 | 11.7 | 37.6 | 51.1 | 40.1 | 37.8 | 51.3 | 39.7 |
> | | SFT-Multi (52k, 1 epoch) | 73.4 | 16.2 | 11.9 | 39.9 | 48.7 | 38.7 | 38.2 | 51.2 | 39.7 |
> | | **SPIRAL-Multi** | **78.2** | **19.7** | 13.3 | **41.8** | **61.6** | **42.6** | **40.1** | **58.5** | **44.5** |
> | **Qwen3-8B**| SFT-Multi (25k, 2 epochs) | 82.8 | 19.9 | 15.6 | 45.9 | 63.5 | 40.8 | 41.6 | 58.8 | 46.1 |
> | | SFT-Multi (52k, 1 epoch) | 81.8 | 22.6 | 14.0 | 48.8 | 61.6 | 40.0 | 43.1 | 57.4 | 46.2 |
> | | **SPIRAL-Multi** | **86.6** | **26.2** | **16.8** | **49.6** | **65.2** | **46.3** | **44.6** | **61.1** | **49.6** |
> | **Octothinker-8B** | SFT-Multi (25k, 2 epochs) | 66.0 | 3.3 | 3.8 | 23.9 | 31.0 | 23.8 | 24.9 | 39.1 | 27.0 |
> | | SFT-Multi (52k, 1 epoch) | 66.4 | 4.2 | 3.6 | 21.9 | 31.6 | 24.6 | 24.9 | 40.8 | 27.3 |
> | | **SPIRAL-Multi** | **68.6** | **5.3** | **4.8** | **33.7** | **43.2** | **32.0** | **33.8** | **49.3** | **33.8** |
> | **Llama-3.1-8B-Instruct** | SFT-Multi (25k, 2 epochs) | **51.8** | 4.6 | 0.7 | 19.1 | 23.3 | 21.7 | 30.0 | 48.9 | 25.0 |
> | | SFT-Multi (52k, 1 epoch) | 51.0 | 4.3 | 0.0 | **21.5** | 23.2 | 21.1 | 30.2 | 49.1 | 25.1 |
> | | **SPIRAL-Multi** | 49.8 | **4.9** | **1.8** | 17.3 | **26.0** | **24.6** | **32.2** | **50.4** | **25.9** |
>
> Doubling the SFT data (successful trajectories from multiple experts) shows no major improvement, while SPIRAL consistently outperforms both baselines.
>
>
> **W3: LLM-as-judge reliability**
>
> To clarify: we evaluate each trace independently (not in batches), use structured prompts with clear pattern definitions, and manually verified samples for agreement. Importantly, we use LLM-as-judge as a post-hoc analysis tool to gain insights into why SPIRAL works, not as a methodological contribution. Pattern statistics are averaged over thousands of traces, reducing noise.

---

> > ### Author Response · Authors · 2025-11-26
> > **Response to Reviewer 9i3G [2/3]**
> >
> > **Main Questions:**
> >
> > **Q3: Model sourcing reasoning traces**
> >
> > Qwen3-4B-Base + SPIRAL-Multi sourced the traces to track pattern evolution. We'll clarify this in Section 4.1.
> >
> > **Q4: Random and RL baselines in Tables 4-5**
> >
> > Following your suggestion, we added Random Policy and Instruct Model baselines for the win-rate against fixed model (Gemini-2.0-Flash) in Table 5:
> >
> > | **Model** | **TicTacToe** (Train) | **KuhnPoker** (Train) | **Simple Neg.** (Train) | **Snake** (OOD) | **Pig Dice** (OOD) | **Truth & Dec.** (OOD) | **Average** |
> > | :--- | :---: | :---: | :---: | :---: | :---: | :---: | :---: |
> > | Base Model | 17.5% | 21.5% | 15.6% | 7.8% | 0.2% | 49.6% | 18.7% |
> > | Random Policy | 24.5% | 31.3% | 8.2% | N/A | N/A | N/A | N/A |
> > | Instruct Model | 52.7% | 48.5% | **46.2%** | 25.2% | 97.6% | 75.5% | 57.6% |
> > | **Multi-Game Model** | 54.3% | **53.9%** | 33.2% | **31.6%** | **99.8%** | **84.0%** | **59.5%** |
> >
> > Our **Multi-Game Model** achieves the highest average win rate (**59.5%**), outperforming the strong Instruct Model baseline (**57.6%**). While the Single-Game Specialist performs best on *Simple Negotiation*, the Multi-Game Model demonstrates superior generalization across the broader suite of training and OOD games. The gap in Simple Negotiation suggests interesting avenues for future work in balancing multi-objective optimization within our self-play paradigm.
> >
> > **Regarding Table 4:** Regarding the request to add these baselines to **Table 4**: We respectfully clarify that the primary objective of Table 4 is a **head-to-head comparison of specialist transferability**. The goal is to isolate how specific learned capabilities (e.g., spatial, probabilistic, or strategic reasoning) transfer between different specialist models. Introducing random or generalist policies to this specific matrix would obscure the focus on relative transfer performance between the specialists. Therefore, we have prioritized the inclusion of these important baselines in Table 5, where they provide the most value in contextualizing overall performance and generalization.
> >
> > **Q5: Framework differences from Nemo RL**
> >
> > Our framework specifically handles multi-agent, multi-turn self-play with shared parameters. Unlike Nemo RL (single-agent), we manage alternating turns, zero-sum dynamics, and role-conditioned training enabling stable self-play for reasoning development.
> >
> > **Q6: Generalization to increased complexity**
> >
> > Great suggestion! We're adding experiments on 5x5 TicTacToe and 5-card Kuhn Poker to assess scalability.
> >
> > | Environment | Setting | Qwen3-4B Base | SFT | SPIRAL (Ours) |
> > | :--- | :---: | :---: | :---: | :---: |
> > | **Tic-Tac-Toe** | 3×3 (Train) | 17.5 | 46.9 | 54.3 |
> > |  | 5×5 (OOD) | 5.5 | 17.8 | 27.3 |
> > | **Kuhn Poker** | 3-Card (Train) | 21.5 | 43.6 | 53.9 |
> > |  | 5-Card (OOD) | 21.9 | 28.6 | 50.1 |
> > | **Simple Negotiation** | 5 Resources (Train) | 15.6 | 26.7 | 33.2 |
> > |  | 8 Resources (OOD) | 5.5 | 8.7 | 31.0 |
> > | **Average (Train)** | | 18.2|	39.1|	47.1
> > | **Average (OOD)** | | 11.0	| 18.4 |	36.1 |
> >
> > The table shows although SFT has comparable  improvement over the base model in in-domain environments (averaging $39.1\%$ vs. $18.2\%$), it struggles to adapt to OOD cases with increased complexity. The SFT model's performance drops  to only $18.4\%$. In contrast, SPIRAL demonstrates better generalizability. It not only outperforms SFT in the training settings (averaging $47.1\%$) but maintains robust performance in the more complex OOD environments, achieving an average of $36.1\%$. This suggests that SPIRAL is more capable of handling increased complexity.
> >
> > **Secondary Questions:**
> >
> > **Q7: Game trajectory statistics**
> >
> > We added comprehensive table with average number of game length, reasoning tokens per step, P1/P2 self-play win rates, and average win-rate and win-rate per game against Gemini-2.0-Flash.
> >
> > | Training Checkpoint | Avg. Game Round (Moves) | Avg. Reasoning Tokens / Step | Self-Play P1 Win-Rate| Self-Play P2 Win-Rate | Avg. Win-Rate vs. Gemini | Win-Rate vs. Gemini (TicTacToe) | Win-Rate vs. Gemini (KhunPoker) | Win-Rate vs. Gemini (SimpleNegotiation) |
> > | :--- | :---: | :---: | :---: | :---: | :---: | :---: |:---: |:---: |
> > | **Step 0** | 1.69 | 4061 | 42.7% | 57.3% | 12.5% | 12.5% | 6.25% | 18.8% |
> > | **Step 128** | 7.63 | 1609 | 48.7% | 51.3% | 28.5% | 16.7% | 37.5% | 31.2% |
> > | **Step 256** | 8.47 | 1755 | 53.2% | 46.8% | 51.8% | 42.9% | 68.8% | 43.8% |
> > | **Step 384** | 9.43 | 1921 | 59.1% | 40.9% | 66.7% | 75.0% | 68.8% | 56.3% |
> > | **Step 400** | 9.55 | 2032 | 62.4% | 37.6% | 67.4% | 83.3% | 62.5% | 56.3% |

---

> > > ### Author Response · Authors · 2025-11-26
> > > **Response to Reviewer 9i3G [3/3]**
> > >
> > > **Q8: Transferable patterns methodology**
> > >
> > > We use GPT-4.1 to classify reasoning traces using structured prompts (Appendix F). We'll add clearer pointers and summarize the methodology in Section 4.
> > >
> > > **Q9: $\tau$ notation and input structure**
> > >
> > > $\tau$ denotes a complete trajectory from initial state to terminal state, containing all state-action pairs $\{(s_0, a_0), (s_1, a_1), ..., (s_T, a_T)\}$. At each turn, the model input consists of: current state $s_t$, player role $p$, and game identifier $G_i$. The model generates: `[reasoning process]\boxed{[action]}`. During policy gradient training, we compute gradients on the entire generated sequence (both reasoning and action components) using REINFORCE with the game outcome as the reward signal. Will clarify these details in Algorithm 1.
> > >
> > > **Q10: GPU hours and compute budget**
> > >
> > > Approximately 25 hours on 8 H100s per experiment for Qwen3-4B; 16 hours on 8 H100s per experiment for Llama3.1-8B-Instruct and Octothinker-8B; 28 hours on 8 H100s for Qwen3-8B.
> > >
> > > **Q11: Game choice justification**
> > >
> > > Games chosen for cognitive diversity: TicTacToe (spatial reasoning), Kuhn Poker (probabilistic reasoning), Simple Negotiation (strategic optimization). OOD games test transfer within these cognitive categories. 400 steps chosen based on convergence analysis. Will add justification section.

---

### Official Review · Reviewer_TDEZ · 2025-11-01

**Soundness:** 3
**Presentation:** 3
**Contribution:** 3
**Rating:** 6
**Confidence:** 4

**Summary:**

This paper presents SPIRAL, a self-play reinforcement learning framework where LLMs learn reasoning by playing multi-turn zero-sum games against themselves. It introduces Role-conditioned Advantage Estimation (RAE) to stabilize multi-agent training. Experiment results show that SPIRAL improves models' performance on reasoning benchmarks like math and reasoning.

**Strengths:**

1. Solid idea and algorithm design: Using self-play to construct an automatic curriculum for improving LLMs’ reasoning ability is a reasonable idea, and the proposed multi-turn, multi-agent RL framework is a novel approach.
2. Good empirical results: The method is thoroughly evaluated across multiple models and benchmarks, showing consistent and meaningful performance improvements.
3. Good clarity and presentation: The manuscript is well written and easy to follow. Figures and tables effectively illustrate both the proposed method and the experimental outcomes.

**Weaknesses:**

1. The second contribution (RAE) is relatively weak: Normalizing advantages separately for different agents in multi-agent settings is a common practice in classical MARL [1, 2]. Applying it to LLM-based multi-agent systems is quite straightforward, which makes the second contribution less substantial.
2. Limited performance on instruct models: In Table 1, SPIRAL shows a significant improvement on base models but much smaller gains (1–2%) on instruct models. This raises the question of whether the improvement on base models truly comes from enhanced reasoning ability or merely better instruction-following.
3. Missing related work: For LLMs in gaming, the authors do not discuss closely related work on LLM agents in multi-agent games such as Diplomacy [3], Werewolf [4, 5], and Avalon [6]. Some of these works also use self-play with RL to train agents, and should be referenced.

[1] Lowe, Ryan, et al. “Multi-Agent Actor-Critic for Mixed Cooperative-Competitive Environments.” arXiv preprint arXiv:1706.02275 (2017).

[2] Yu, Chao, et al. “The Surprising Effectiveness of PPO in Cooperative Multi-Agent Games.” Advances in Neural Information Processing Systems (2022).

[3] Bakhtin, Anton, et al. “Human-Level Play in the Game of Diplomacy by Combining Language Models with Strategic Reasoning.” Science 378.6624 (2022): eade9097.

[4] Xu, Yuzhuang, et al. “Exploring Large Language Models for Communication Games: An Empirical Study on Werewolf.” arXiv preprint arXiv:2309.04658 (2023).

[5] Xu, Zelai, et al. “Language Agents with Reinforcement Learning for Strategic Play in the Werewolf Game.” arXiv preprint arXiv:2310.18940 (2023).

[6] Wang, Shenzhi, et al. “Avalon’s Game of Thoughts: Battle Against Deception through Recursive Contemplation.” arXiv preprint arXiv:2310.01320 (2023).

**Questions:**

compare them with the corresponding base model, to better analyze how much of the improvement on base models comes from genuine reasoning enhancement?
2. Please analyze why training on games improves QA benchmarks. The paper provides detailed analysis on why SPIRAL enhances performance on math benchmarks, but the improvement on QA seems less straightforward and lacks detailed explanation.
3. During training, the base model cannot directly use the <think> special token, but the pattern analysis section mentions the “thinking token.” How was this implemented in practice—was it through CoT prompting or another mechanism?

---

> ### Author Response · Authors · 2025-11-26
> **Response to Reviewer TDEZ [1/2]**
>
> Thank you for reviewing our work carefully and for recognizing it as a "novel approach" with "solid idea and algorithm design". We appreciate your valuable feedback and have addressed your concerns with new experiments and clarifications.
>
> **W1: RAE contribution and thinking collapse prevention**
>
> Our ablation study (Figure 6) demonstrates RAE's critical importance: without RAE, reasoning traces catastrophically collapse from 1,500 to near zero characters after 200 steps, with math performance dropping 66% (from 35% to 12%). This complete abandonment of chain-of-thought reasoning represents a unique failure mode in LLM self-play that hasn't been observed in traditional MARL. This brings the key insight why RAE is more important in the LLM self-play settings, as part of our contribution.
>
> **W2: Qwen3 Instruct Results**
>
> We thank the reviewer for the insight. Please note that Llama-3-instruct itself is a rare weak reasoning model, which limits its upper bound on learning better reasoning skills and limited its performance gain on math benchmark. To better study our method effectiveness, we perform additional experiments with **Qwen3-4B-Instruct-2507**.
>
> As shown in the table below, standard SFT on reasoning trajectories slightly degrading performance. In contrast, **SPIRAL** successfully reverses this trend. SPIRAL achieves a **2\% improvement** on over the original model and a substantial average gain of **3.6%** over the SFT baseline, reaching a total average score of **75.51%**. This demonstrates that self-play can refine genuine reasoning capabilities even in models that have strong performance.
>
> | Model | Math500 | AIME24 | AIME25 | Olympiad | AMC-23 | Minerva | GPQA-D | MMLU-Pro | Average |
> | :--- | :---: | :---: | :---: | :---: | :---: | :---: | :---: | :---: | :---: |
> | **Qwen3-4B-Instruct** (Base) | 91.2 | 64.6 | 47.4 | **89.6** | 82.1 | **87.6** | **62.0** | 69.6 | **74.10** |
> | + SFT-Multi | 90.8 | 59.3 | 46.1 | 86.4 | 79.6 | 82.3 | 61.4 | 69.1 | **71.88** |
> | **+ SPIRAL-Multi (Ours)** | **93.5** | **67.9** | **49.8** | 88.8 | **83.3** | 85.5 | 61.8 | **71.5** | **75.91** |
>
> **W3: Discussion of Related Work (Diplomacy, Werewolf, Avalon)**
>
> Thank you for these valuable references. We will incorporate [3-6] into our related work section to better contextualize SPIRAL. While we share the use of games as a testbed, there are fundamental distinctions in **objective** and **architecture** between these works and SPIRAL:
>
> * **Main difference: Game Mastery vs. Cognitive Transfer:** The primary objective of works such as *Cicero* (Diplomacy) [3], *Werewolf* agents [4,5], and *Avalon* [6] is **in-domain mastery**, optimizing the agent to win a specific game through strategic communication and deception. For instance, Cicero [3] combines a language model with a separate strategic planning algorithm specifically to reach human-level performance in Diplomacy. In contrast, SPIRAL treats games not as the end goal, but as a **training scaffold**. Our objective is not to build the best Poker player, but to use the pressure of zero-sum competition to induce generalizable reasoning patterns (like case-by-case analysis) that transfer to **out-of-domain** academic benchmarks (Math, GPQA). We believe this focus on *transfer* rather than *gameplay* sets SPIRAL apart.

---

> ### Author Response · Authors · 2025-11-26
> **Response to Reviewer TDEZ [2/2]**
>
> **Q2: Games improving QA benchmarks with concrete examples**
>
> Following your suggestion, we analyzed reasoning pattern transfer using our systematic framework (Appendix F). Table 2 shows three core patterns that emerge during gameplay and transfer to academic reasoning:
>
> In Kuhn Poker, models develop **Case-by-Case Analysis**: "Case 1 - Fold: You lose 1 chip. You have 2 chips now. Case 2 - Call: You have 0% chance of winning, so you will have 1 chip if you lose. Since losing 1 chip (fold) is better than losing 2 chips (call and lose), the best action is to fold. $\boxed{\text{fold}}$"
>
> This systematic enumeration transfers directly to QA problems. For instance, in Minerva Problem 135 (Appendix A.1), the model progresses from unstructured reasoning at checkpoint 16 to systematic case-by-case analysis by checkpoint 400: "Case 1 - Subproblem 0: Understand Context: He formation in early universe... Round to Power of 10: $\boxed{20}$ minutes." The same structured approach that evaluates game options now decomposes complex scientific questions.
>
> Figure 4 quantifies this transfer across 290 game trajectories and 46,792 math solutions: Case-by-Case Analysis transfers at 72%, Pattern Recognition amplifies from 35% to 45%, while Expected Value Calculation selectively transfers at 28% where probabilistic reasoning applies. MMLU-Pro benefits from systematic knowledge application across disciplines using these game-learned decomposition strategies. We've detailed these transfer mechanisms in Section 4 (lines 423-438).
>
> **Q3: Thinking tokens implementation**
>
> Thanks for catching this! You're right about the notation. During training and inference, the model generates structured outputs in the format: `[reasoning process]\boxed{[action]}`. This structured generation approach (not special tokens) enables chain-of-thought reasoning. The `<output></output>` tags in our paper are only used to indicate model-generated content for clarity in examples, not as part of the actual training format. We'll correct this notation in Section 3 and ensure consistency throughout.

---

### Author Response · Authors · 2025-11-26
**General Response to Reviewers**

We want to thank all four reviewers for their time and their detailed, constructive feedback. We are very encouraged that the reviewers converged on several key strengths of our work.

* **Novel Self-Play Framework:** Reviewers consistently praised the innovation of using multi-agent self-play to generate an "autonomous curriculum" (**2RMj**). Reviewer **TDEZ** commended the "solid idea and algorithm design," while **Q6Ug** highlighted that this "novel and compelling framework" makes a "major contribution" to the field by reducing dependence on human-curated data.
* **Strong Empirical Results:** The evaluation was recognized as "comprehensive" (**Q6Ug**) and "thorough" (**TDEZ**). Reviewers highlighted the "consistent and meaningful performance improvements" (**TDEZ**) and "substantial gains" (**2RMj**, **9i3G**) across diverse model families. Reviewer **2RMj** specifically noted the "broad applicability" demonstrated across base, instruct, and RLVR-trained models.
* **Mechanistic Insights on Transfer:** Reviewers appreciated that our analysis went beyond benchmark scores to explain *why* the method works. Reviewer **Q6Ug** praised the "deep and insightful analysis of skill transfer," and **2RMj** noted that our approach to tracking specific reasoning patterns provided valuable "mechanistic insights." Reviewer **9i3G** added that our automatic evaluation methodology could serve as a "blueprint" for future research.

**Common Questions**

We appreciate this positive feedback and have addressed the primary common concerns regarding experimental robustness and baselines with extensive new experiments.

### 1. Statistical Robustness & Reproducibility (Response to 9i3G)
To address concerns regarding statistical significance, we re-ran our main experiments with **3 random seeds** (seeds 14, 42, 100). As shown below, SPIRAL consistently outperforms the SFT baseline with narrow confidence intervals, confirming the robustness of our gains.

| Model | Math500 | AIME24 | AIME25 | Olympiad | Average |
| :--- | :--- | :--- | :--- | :--- | :--- |
| **Qwen3-4B-Base** | 73.4 | 9.6 | 6.2 | 33.3 | 34.0 |
| **SFT-Multi** | $74.0 \pm 0.6$ | $12.4 \pm 1.5$ | $11.2 \pm 1.1$ | $37.8 \pm 0.3$ | $39.6 \pm 0.4$ |
| **SPIRAL-Multi (Ours)** | $\mathbf{78.7 \pm 2.0}$ | $\mathbf{18.8 \pm 2.5}$ | $\mathbf{15.0 \pm 1.3}$ | $\mathbf{41.8 \pm 1.3}$ | $\mathbf{44.5 \pm 0.5}$ |

### 2. Stronger SFT Baselines (Response to 9i3G)
Reviewer **9i3G** suggested the SFT baseline might be underpowered. We generated a massive dataset of **51,200 trajectories** (matching SPIRAL's 400×128 scale) from multiple expert models. Doubling the SFT data did not yield significant gains, whereas SPIRAL continued to outperform both the 25k and 52k SFT baselines. This confirms that SPIRAL’s improvement stems from the **adaptive curriculum**, not just data volume.

| Model | Method | Math500 | AIME24 | Olympiad | MMLU-Pro | Avg |
| :--- | :--- | :--- | :--- | :--- | :--- | :--- |
| **Qwen3-4B** | SFT-Multi (25k) | 74.2 | 13.7 | 37.6 | 51.3 | 39.7 |
| | SFT-Multi (52k) | 73.4 | 16.2 | 39.9 | 51.2 | 39.7 |
| | **SPIRAL-Multi** | **78.2** | **19.7** | **41.8** | **58.5** | **44.5** |

### 3. General Capabilities & Pipeline Integration (Response to 2RMj & TDEZ)
We addressed concerns about "catastrophic forgetting" and SPIRAL's role in the training pipeline.
* **No Forgetting:** On **AlpacaEval-2**, SPIRAL improves win rates significantly over Base models (e.g., Qwen3-4B: $6.98\% \to \mathbf{27.6\%}$) and matches or exceeds SFT. **IFEval** accuracy also improves ($37.3\% \to \mathbf{46.4\%}$), confirming instruction following is preserved.
* **Pipeline Booster:** We compared `Base -> SPIRAL -> RLVR` vs. `Base -> RLVR` using Math-12k. SPIRAL acts as a complementary booster:
    * Base $\to$ RLVR: Avg Score **46.0**
    * Base $\to$ SPIRAL $\to$ RLVR: Avg Score **46.5**
    * Base $\to$ RLVR $\to$ SPIRAL: Avg Score **48.1**

### 4. Generalization to Complexity (Response to 9i3G)
We tested SPIRAL on OOD environments with increased complexity (5x5 TicTacToe, 5-Card Kuhn Poker). SPIRAL achieved a **36.1% OOD win rate** compared to SFT's **18.4%**, demonstrating that self-play fosters genuine strategic generalization rather than memorization.

We thank the reviewers for these suggestions, which have significantly strengthened the empirical rigor of our paper. We address specific questions in the individual threads below.

---

### Author Response · Authors · 2025-12-03
**Summary of the Update and New Revision of our Paper**

Based on the reviewers' feedback, we have updated our manuscripts accordingly and uploaded a new version of our paper for review. The changes are highlighted in blue. We summarize the key changes:

* **Added multi-seeded experimental results** in **Section 4** and **Table 1**, reporting mean $\pm$ standard deviation across 3 random seeds to demonstrate statistical robustness (Reviewer 9i3G).

* **Incorporated a stronger SFT baseline** (52k trajectories) in **Appendix**, verifying that SPIRAL's gains stem from the adaptive curriculum rather than simply scaling dataset size (Reviewer 9i3G).

* **Added pipeline integration experiments** in **Appendix**, comparing `Base -> SPIRAL -> RLVR` vs `Base -> RLVR` to demonstrate SPIRAL's effectiveness as a mid-training booster (Reviewer 2RMj).

* **Included Qwen3-Instruct experiments** in **Appendix**, showing SPIRAL reverses the performance regression seen with standard SFT on instruction-tuned models (Reviewer TDEZ).

* **Added General Capabilities evaluation** (AlpacaEval-2, IFEval) in **Appendix**, confirming that SPIRAL preserves instruction-following abilities while improving reasoning (Reviewer 2RMj).

* **Expanded the Generalization analysis** in **Appendix**, adding experiments on increased complexity environments (e.g., 5x5 TicTacToe, 5-card Kuhn Poker) to assess scalability (Reviewer 9i3G).

* **Expanded the Related Work** in **Section 2** to discuss multi-agent gaming works (Diplomacy, Werewolf, Avalon), distinguishing their focus on game mastery from SPIRAL's focus on cognitive transfer (Reviewer TDEZ).

* **Clarified Algorithm 1 and notation** in **Section 3**, explicitly defining trajectory $\tau$, input structures, and correcting the "thinking token" notation to reflect the structured output format (Reviewer TDEZ, 9i3G).

* **Added Game Choice Justification** in **Section 4**, detailing the specific cognitive skills targeted by each game (spatial, probabilistic, strategic) and justifying the training steps (Reviewer 2RMj).

* **Clarified the failure of fixed opponents** in **Section 4**, explaining why static baselines (even with positive expected value) fail to drive reasoning improvement compared to adaptive self-play (Reviewer Q6Ug).

* **Added Compute Resource details** in **Section 4**, specifying the GPU hours required for different model scales to ensure reproducibility (Reviewers 9i3G, Q6Ug).

* **Added Game Trajectory Statistics** in **Appendix**, detailing game lengths, reasoning tokens, and win rates across training checkpoints (Reviewer 9i3G).

---

### Meta-Review · Area_Chair_EoE3 · 2025-12-27

**Summary:**

The paper introduces SPIRAL, a self-play reinforcement learning framework where LLMs develop transferable reasoning skills by playing multi-turn zero-sum games. The authors propose Role-conditioned Advantage Estimation (RAE) to stabilize multi-agent training and prevent reasoning trace collapse. While the framework demonstrates significant gains across multiple reasoning benchmarks (up to 10%) without requiring human-curated problem-answer pairs, the core discussion during the review phase focused on the method's positioning relative to existing RLVR pipelines.

**Reviewer Concerns:**

Reviewer Concerns
- **Addressed Concerns** (Reviewer 9i3G): The authors successfully addressed concerns regarding statistical robustness and baseline strength by providing multi-seeded experimental results (showing narrow confidence intervals) and results from a massive 52k-trajectory SFT baseline.
- **Addressed Concerns** (Reviewer TDEZ): Concerns regarding performance on instruction-tuned models were addressed with new Qwen3-Instruct experiments, showing SPIRAL reverses typical SFT performance regressions.
- **Outstanding Concerns** (Reviewer 2RMj): While new pipeline experiments (Base $\rightarrow$ SPIRAL $\rightarrow$ RLVR) show SPIRAL acts as a complementary "booster," the claim that it can fully replace domain-specific human-curated data remains somewhat overclaimed. The results suggest SPIRAL is an effective mid-training or post-RLVR stage rather than a complete alternative to verifiable reward training8.

**Reviewer Scores:**

- Reviewer Q6Ug (Score 8): Would likely maintain their score as the rebuttal reaffirmed the major findings and addressed their concerns.
- Reviewer TDEZ (Score 6): Would likely at least maintain their score given the substantial gains (3.6%) demonstrated on the requested Qwen3-Instruct models.
- Reviewer 9i3G (Score 4): This score would likely improve to a 6, as the authors provided all requested statistical rigor, multi-seeded evaluations, and computational budget details.
- Reviewer 2RMj (Score 4): This review should be weighed lightly due to several factual hallucinations identified by the authors (e.g., claiming missing experiments that were present); however, their concern regarding overclaiming the replacement of RLVR remains a valid point for the final revision.

---

### Decision · Program_Chairs · 2026-01-26

Accept (Poster)